# The dual coding of a single sex pheromone receptor in Asian honeybee *Apis cerana*
Haoqin Ke[1,4], Jonathan D. Bohbot[2,4], Yongjuan Chi[3], Shiwen Duan[1], Xiaomei Ma[1], Bingzhong Ren[1] ✉ & Yinliang Wang ● [1] ✉

In Asian honeybees, virgin queens typically only mate during a single nuptial flight before founding a colony. This behavior is controlled by the queen-released mandibular pheromone (QMP). 9-oxo-(*E*)-2-decenoic acid (9-ODA), a key QMP component, acts as sex pheromone and attracts drones. However, how the queens prevent additional mating remains elusive. Here, we show that the secondary QMP component methyl *p*-hydroxybenzoate (HOB) released by mated queens inhibits male attraction to 9-ODA. Results from electrophysiology and in situ hybridization assay indicated that HOB alone significantly reduces the spontaneous spike activity of 9-ODA-sensitive neurons, and *AcerOr11* is specifically expressed in sensilla placodea from the drone's antennae, which are the sensilla that narrowly respond to both 9-ODA and HOB. Deorphanization of *AcerOr11* in *Xenopus* oocyte system showed 9-ODA induces robust inward (regular) currents, while HOB induces inverse currents in a dose-dependent manner. This suggests that HOB potentially acts as an inverse agonist against *AcerOr11*.

A Virgin honeybee queen usually mates only once with several drones, and for the rest of her prolific life, she will not engage in subsequent mating events[1,2]. The queen mandibular pheromone (QMP) plays a key role in regulating colony reproduction. QMP directly triggers the mating behavior and provides information about the mating status of queens. It also inhibits the development of worker ovaries by changing relevant gene expression[3]. Several studies have focused on *Apis mellifera* QMP, which consists of four main components: 9-oxo-(*E*)-2-decenoic acid (9-ODA), (*R*, *S*)-9-hydroxy-(*E*)-2-decenoic acid (9-HDA), methyl *p*-hydroxybenzoate (HOB), and 4-hydroxy-3-methoxyphenyl-ethanol (HVA)[4]. To date, only 9-ODA has been shown to have an attractive effect on drones[3]. In *A. cerana*, 9-ODA, 9-HDA, and HOB are also the dominant component of QMP, while HVA is absent[4,5]. Behavioral evidence suggests that the QMP mixture lacking HVA in *A. cerana* is sufficient to elicit retinue behavior in workers[4], indicating a similar function of QMP in *A. cerana* to that in *A. mellifera*. However, the role of HOB remains poorly understood, as single secondary QMP components are not attractive to drones[6,7]. So far, the mechanism by which HOB alone regulates mating behavior in *A. cerana* has remained unknown.

Olfactory sensing of QMP has been extensively studied in *A. mellifera*. At the peripheral olfactory system level, the primary QMP component, 9-ODA, is detected by the placoid sensilla located on the drone antenna, which expresses the *A. mellifera* odorant receptor 11 gene (*AmelOr11*)[8]. When expressed in the *Xenopus laevis* system, *AmelOr11* exhibits a narrow response to 9-ODA and does not respond to other QMP components[9]. This suggests the involvement of other olfactory proteins in detecting QMP secondary components. The predominant role of 9-ODA in mating is further supported by calcium imaging experiments at the antennal lobe level, where the *A. mellifera* macroglomerulus MG2 in drones is activated by 9-ODA[10]. Although *A. mellifera* and *A. cerana* share overall similar morphology and social behavior, they differ in their olfactory systems, as well as QMP composition, the number of odorant receptors (Ors), and antennal lobe topology[11]. The QMP olfactory sensing in *A. cerana* has received less attention compared to *A. mellifera*. In *A. cerana*, odorant binding protein 11 (AcerOBP11) demonstrates strong binding affinities for both 9-ODA and HOB[12]. However, the specific ORs responsible for detecting 9-ODA and HOB in *A. cerana* remain unclear. Our transcriptome data showed that the 9-ODA receptor ortholog also exists in *A. cerana*, which shed a light on discovery the olfactory pathway on sex pheromone sensing in Asian honeybees[13].

In this study, we found that HOB, released only by mated queens, significantly reduces the attraction of drones to 9-ODA. This inverse effect of HOB was further validated by in vivo electrophysiological assays, electroantennography (EAG), and single sensillum recording (SSR). Lastly, we uncovered that, similar to the *A. mellifera* orthologs, *AcerOr11* is robustly

[1]Key Laboratory of Vegetation Ecology, MOE, Northeast Normal University, Changchun, China. [2]Department of Entomology, The Hebrew University of Jerusalem, The Robert H. Smith Faculty of Agriculture, Food and Environment, Rehovot, Israel. [3]Apiculture Science Institute of Jilin Province, Jilin, China. [4]These authors contributed equally: Haoqin Ke, Jonathan D. Bohbot. ✉e-mail: bzren@nenu.edu.cn; wangyl392@nenu.edu.cn

activated by 9-ODA, moreover, the secondary component HOB elicited reverse current fluxes, implying the existence of a dual coding mechanism at the QMP receptor, *AcerOr11*. This study aimed to explore how QMP regulates reproduction at the olfactory sensing level of *A. cerana*.

## Results

### Queen-released HOB reduces 9-ODA attraction to drones

To measure the contents of QMPs in *A. cerana*, we performed GC-MS analysis on 12- to 15-day-old virgin queens, mated queens as well as drones and workers. 9-ODA and 9-HDA, two main components in QMPs, were detectable in both virgin and mated queens (Fig. 1A, B, Supplementary Fig. 1). However, interestingly, HOB was only detected in the mated queens and not in virgin queens (Fig. 1B). This inspired us that HOB might exercise some functions on the post-mating regulation. Notably, the amount of HOB released by mated queens was significantly smaller than that of 9-ODA (Supplementary Table 1).

Thus, we aimed to test the behavioral effects of HOB on drones, using the Y-tube olfactometer assay. First, we used 9-ODA as a stimulus and found that it has a significant attractive effect on drones only at the highest concentration (100 µg) ($p < 0.05$, two-tailed, T-test). At a lower concentration of 0.1–10 µg, the attraction effect of 9-ODA was not significant ($p = 0.5614$ for 0.1 µg; $p = 0.9580$ for 1 µg; $p = 0.0668$ for 10 µg) (Fig. 2A). While HOB alone did not elicit any effect on drones at any of the tested doses (0.1–100 µg) (Fig. 2B). When we mixed 100 µg of 9-ODA with different concentrations of HOB (0.1–100 µg), we found that 9-ODA's attraction was suppressed by HOB at concentration beyond 1 µg ($p = 0.2302$ for 1 µg; $p = 0.3771$ for 10 µg; $p = 0.9414$ for 100 µg) (Fig. 2C). This suggested that queens only released HOB after mating, which compromises the attraction to drones.

### HOB inhibited 9-ODA neurons in sensilla placodes

To explore the physiological role of HOB in vivo, we conducted an EAG assay. First, we tested the olfactory response in the antenna to 9-ODA or HOB alone in different castes. 9-ODA elicited significant EAG responses in drones at 100 µg (Fig. 3A); the normalized EAG response was $64.63 \pm 2.08$, which was nearly 64 times higher than that of the negative control group (paraffin oil). The antenna of workers and queens only exhibited a mild response to 9-ODA ($N = 5$–$9$, $p < 0.05$, Wilcoxon signed-ranked test) (Fig. 3C, Supplementary Fig. 2A, B). On the contrary, HOB did not elicit any significant antennal response in any castes (Fig. 3B and Supplementary Fig. 2C, D).

We next stimulated the antenna with 9-ODA-HOB mixtures. The 9-ODA concentration was fixed to 100 µg for a saturated EAG response. We

observed an inhibitory effect of HOB on the EAG response to 9-ODA (Supplementary Fig. 3) in a dose-dependent manner. For mixtures with 0.1 and 1 µg HOB, the normalized EAG responses to 9-ODA decreased to $33.33 \pm 4.34$ and $20.05 \pm 2.94$, respectively ($N = 6$–$7$, $p < 0.05$, One-way ANOVA followed by Turkey's test), and the effects were significantly lower than those from 9-ODA alone (Fig. 3D). Intriguingly, HOB at 100 µg eliminated the effect of all tested 9-ODA concentrations (1–100 µg; $N = 7$, $p < 0.05$) (Fig. 3E).

To examine the in vivo effect of HOB on the ORN response, we conducted single sensillum recordings (SSRs). Three types of chemosensory sensilla were observed in the *A. cerana* drone's antenna, including the sensilla trichodea, placodea, and basiconica. Sensilla placodea outnumbers the other two types of sensilla. Of over 150 chemosensory sensilla tested with 9-ODA and HOB, only sensilla placodea showed responses to 9-ODA and HOB, and no response were detected in sensilla trichodea and sensilla basiconica. Consistent with *A. mellifera* studies[8], three types of spike amplitudes were observed in *A. cerana* sensilla placodea, suggesting the presence of three ORNs (neurons A, B, and C) (Fig. 4A). When stimulated with 9-ODA, the activity of the A neurons was significantly increased compared to the negative control ($N = 5$, $p < 0.001$, two-tailed, T-test) (Fig. 4B, C), when stimulated with HOB, the spontaneous activity of the A neuron was significantly reduced ($N = 5$, $p < 0.001$). This indicates the inverse effect of HOB. Intriguingly, neither 9-ODA nor HOB changed the spontaneous activity of B and C neuron (Fig. 4D), suggesting that they were specifically acting on A neuron. We next tested the effect of the 9-ODA-HOB mixture on A neurons. The presence of HOB inhibited 9-ODA-induced activity in A neurons ($N = 5$, $p < 0.05$) (Fig. 4E). Taken together, these results indicate that HOB inhibits not only the antennal response to 9-ODA, but also the spontaneous firing in the absence of 9-ODA. Overall, these results implicating that HOB could act as an inverse agonist at the ORNs level.

### *AcerOr11* is abundantly expressed in sensilla placodea

To check the expression profile of *AcerOr11* underlying the physiological response to 9-ODA and HOB in the antenna, we conducted an RT-qPCR survey of the *AmelOr11* ortholog in *A. cerana* in different tissues and castes. We found that *AcerOr11* is abundantly expressed in drone antennae, while little expression was also detected in the queen and worker antennae (Supplementary Fig. 4).

To further determine the expression level and location of *AcerOr11*, we conducted in situ hybridization experiment. The DIG-labeled riboprobes

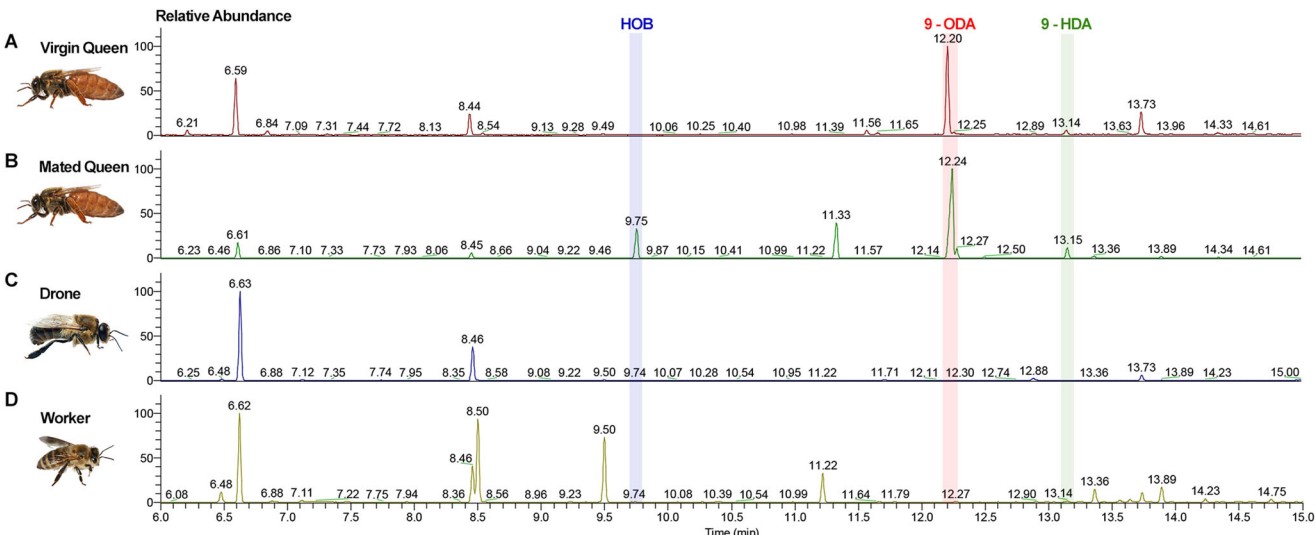

**Fig. 1 | GC-MS analysis of *A. cerana* head extracts.** GC-MS analysis of head extracts from **A** 12- to 15-day-old virgin queens, **B** mated queens (mated on day 6 or 7), **C** 12- to 15-day-old drones, and **D** 12- to 15-day-old workers.

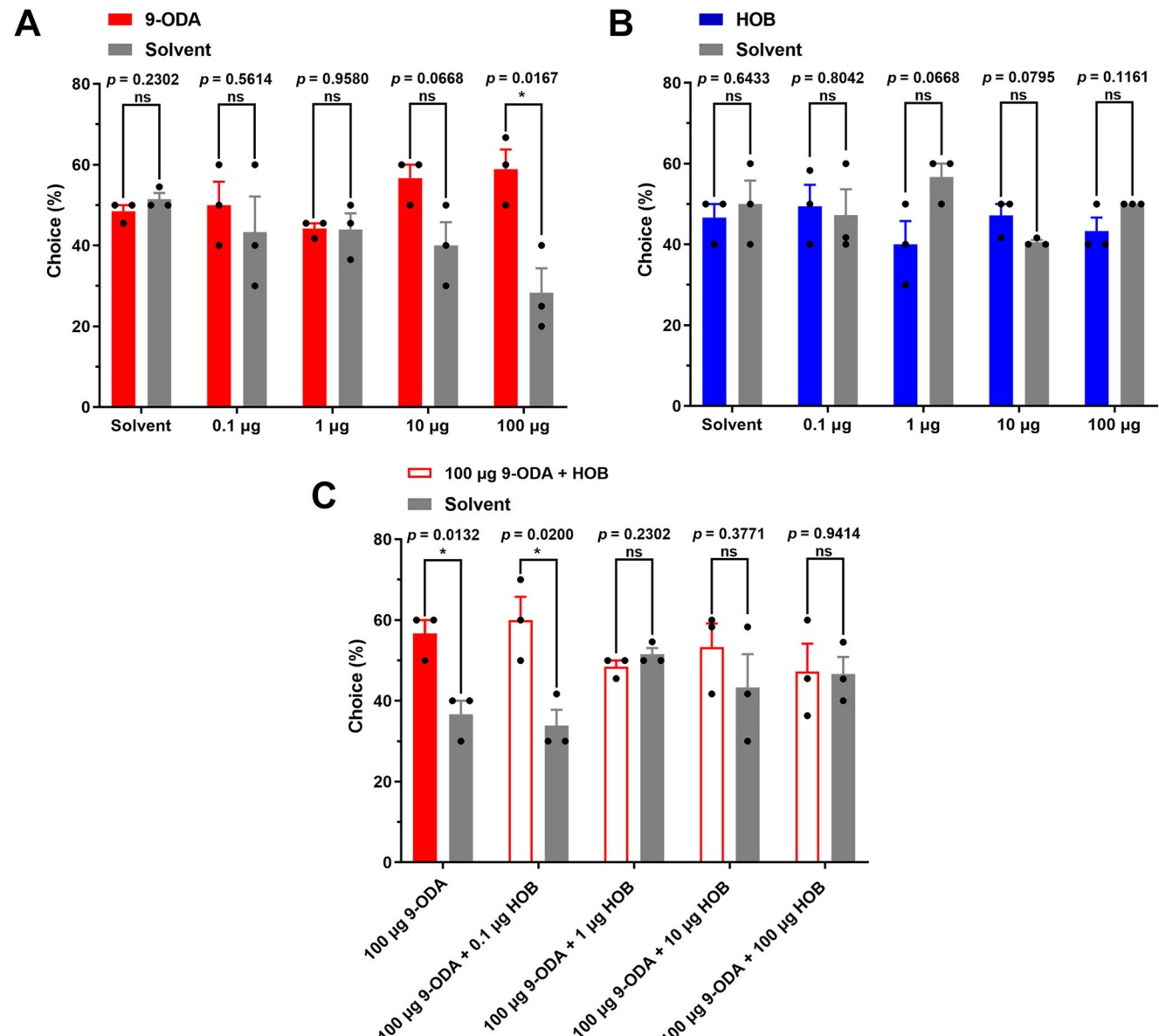

**Fig. 2 | Behavioral responses of *A. cerana* to 9-ODA and HOB.** Behavioral responses of drones to **A** 9-ODA alone, **B** HOB alone, and **C** the mixture ($N = 3$ replicates, 10–12 biologically independent samples per replicate, two-tailed, T-test).

for *AcerOr11* were applied to transversal antennal sections of bees from three castes. We found that many cells in the drone antenna expressed *AcerOr11*, which was uniformly distributed from the F1 to the F11 segments (Fig. 5A, B). Only a few cells in workers' antennae expressed *AcerOr11*, while *AcerOr11* was fully undetectable in queens' antennae (Fig. 5C, D, L, M). No labeled cells were observed in the negative control group (Supplementary Fig. 5). *AcerOr11*-labeled areas were mainly distributed in dendrite-like structures of ORNs housed in sensilla placodea (Fig. 5K). The *AcerOr11* and 4',6-diamidino-2-phenylindole (DAPI) labeled areas were neatly separated (Fig. 5N), suggesting that *AcerOr11* is expressed in the ORN cytoplasm instead of the nucleus.

**Dual coding of *AcerOr11* against 9-ODA and HOB**

To identify the QMP ORs in *A. cerana*, we cloned *AcerOr11*, which is the 1:1 ortholog of *AmelOr11* (9-ODA receptor in *A. mellifera*) and specifically expressed in sensilla placodea from drones[9]. We functionally expressed *AcerOr11* in the *Xenopus laevis* system and screened it with a 163-compound panel. AcerOr11 produced robust regular currents in response to increasing concentrations of 9-ODA ($EC_{50} = 0.35$ nM). Meanwhile, HOB

elicited inverse currents (Supplementary Fig. 6) in a dose-dependent manner ($EC_{50} = 150$ nM) (Fig. 6A–D).

To further confirm the inverse response of *AcerOr11* to HOB, a current-voltage (I–V) curve was constructed using 9-ODA and HOB as stimuli. The results showed that in the range of −80 to +40 mV, the slope of the 9-ODA-elicited I–V curves was much higher (9.729) than that of the baseline (4.015) (Fig. 6E). On the contrary, HOB-generated I–V curves had a much lower slope (2.689) compared with the baseline (4.032) (Fig. 6F). These results validated that the presence of the agonist 9-ODA dramatically increases the conductivity of the cell membrane, and channels are massively opened. Contrarily, the presence of HOB decreases the conductivity as much as in the resting state.

We next measured the TEVC response of the 9-ODA-HOB mixture under different dose ratios. We fixed the concentration of 9-ODA to $10^{-5}$ M for a saturated response while increasing the HOB concentration from $10^{-6}$ M to $10^{-3}$ M. We found a clear dose-dependent inhibitory effect of HOB on 9-ODA activity (Supplementary Fig. 7). At $10^{-4}$ M or higher, HOB significantly inhibited 9-ODA-elicited currents ($N = 4$, $p < 0.05$, One-way ANOVA followed by Turkey's test). At $10^{-3}$ M HOB concentration, the

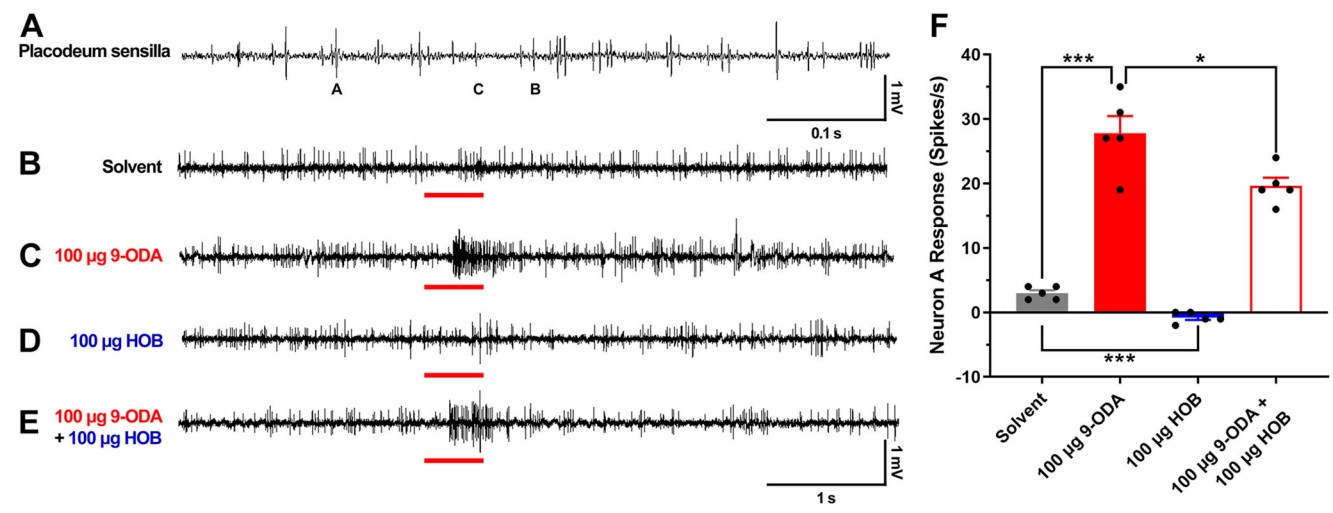

**Fig. 3 | EAG response of 9-ODA and HOB in *A. cerana*. A**, **B** EAG responses of drones to increasing doses (0.01 to 100 µg) of 9-ODA and HOB. **C** Dose-response curves for normalized EAG response to 9-ODA and HOB in *A. cerana* ($N = 5$–9, $p < 0.05$, Wilcoxon signed-ranked test). **D**, **E** HOB elicited dose-dependent inhibition of EAG responses to 9-ODA ($N = 6$–7, $p < 0.05$, One-way ANOVA followed by Turkey's test). Relative EAG responses = Em/CKm, Em represent the mean responses for the test volatile compound and CKm is a negative control.

**Fig. 4 | SSR with 9-ODA and HOB in placodeum sensilla. A–E** Representative traces of placodeum sensilla response to solvent, 100 µg 9-ODA, 100 µg HOB, and the 9-ODA-HOB mixture. **F** Statistical analysis of A neuron's spikes to 100 µg 9-ODA, 100 µg HOB, and the 9-ODA-HOB mixture ($N = 5$, two-tailed, T-test).

AcerOr11 response to the 9-ODA-HOB mixture was 208.4 ± 11.8 nA, which was nearly half of that induced by 9-ODA alone (445.6 ± 29.9 nA). Subsequently, we used 9-ODA alone for a series of concentration tests and found that the response to 9-ODA was recovered (Fig. 6G). These results suggest

that HOB inhibits the Or11 response to 9-ODA. Likewise, when HOB was at $10^{-3}$ M and 9-ODA was at lower doses ($10^{-8}$ M and $10^{-7}$ M), the mixture still induced depolarization (inverse) currents. However, once 9-ODA was raised to $10^{-6}$ M or higher, the response turned to inverse currents (Fig. 6H).

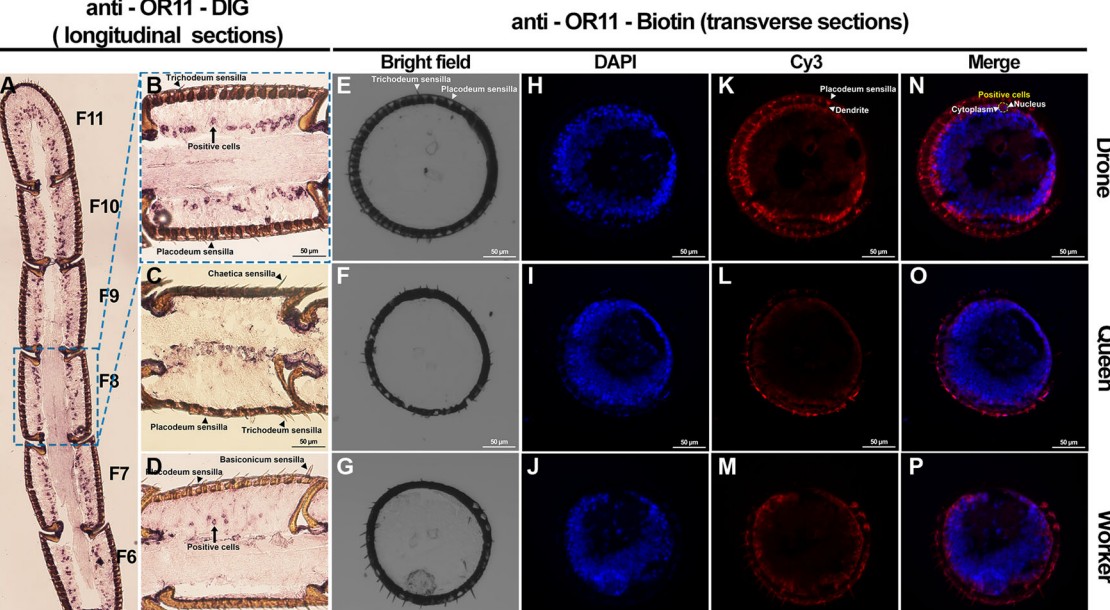

**Fig. 5 | In situ hybridization of *AcerOr11*. A–D** Chromogenic in situ hybridization using the anti-Or11-DIG specific probe on longitudinal sections through the median plane of flagellar segments from antennae of **A**, **B** drones, **C** queens, and **D** workers. **E–P** High magnification images of a two-color FISH experiment on a transverse section incubated with anti-*AcerOr11*-Biotin for *AcerOr11* (**K**, **L**, and **M**; red), and the counterstained images with DAPI (**H**, **I**, and **J**; blue). Bright-field (**E**, **F**, and **G**) and merged (**N**, **O**, and **P**) images are also presented for reference.

## Discussion

HOB, a pheromone signal only released by mated queens, exerts a "stop mating" function on drones. While a few studies have also reported the presence of tiny levels of HOB in virgin queens[14], most of the QMP studies, as well as ours, showed HOB is fully absent in virgin queens[4,15]. To begin with, GC-MS analysis indicated that HOB was only released by mated queens in *A. cerana*, which is consistent with a report about HOB in *A. mellifera* queens[4,7]. In our case, HOB alone did not elicit any behavioral effect on drones, even at the highest doses, in the binary choice assay. However, it significantly reduced the attraction of drones to 9-ODA. Therefore, HOB takes an inverse behavioral effect compared with 9-ODA in *A. cerana*. Previous study showed that the antennal-specific protein 1 (*Asp1*), has a high affinity for HOB, and the *Asp1* expression level is positively correlated with colony sizes in both *A. cerana* and *A. mellifera*[16], these results support that HOB alone could be detected by *Apis* honeybees at a peripheral olfactory sensing level, and might play a crucial role in stop mating in honeybees. Our behavioral assay results indicate that HOB directly inhibits the mating behavior of drones and might prevent the queens from consuming the energy for redundant mating.

In insects, excitatory and inverse chemical signals are equally important for maintaining the population dynamic. In *Helicoverpa armigera*, female moths release (Z)-11-Hexadecen-1-ol (Z11-16: OH) to repel males and avoid non-optimal mating[17]. Likewise, in *Drosophila*, Gr8-associated alkenes inhibit courtship behaviors[18]. Collectively, these findings suggest that the inverse agonist may be widespread in insects. With a few exceptions, honeybee queens do not remate in their lifetime[19]. Mated queens release HOB to prevent drones from multiple mating, which can help in avoiding resource waste due to multiple mating and subsequent breeding.

Interreceptor inhibition by semiochemicals at the sensillum level, a well-established phenomenon in *D. melanogaster*, is mediated by non-synaptic "lateral inhibitions" between neurons located in the same sensilla, termed ephaptic coupling[20,21]. In mosquitoes, high concentrations of ammonia elicit atypical bursts of action potentials, followed by inhibition in multiple adjacent ORNs[22]. In *Culex. quinquefasciatus*, eucalyptol significantly decreases the number of spikes in the ab7 sensillum[23]. Besides the interreceptor inhibition, here, we show that HOB reduces the activity of 9-ODA at the same ORNs, supporting the hypothesis of the intrareceptor inhibition. Notably, HOB alone did not elicit any "normal" EAG response, which negates the possibility of any HOB-activated ORs, i.e., HOB does not act as an agonist. Since HOB reduces spontaneous spikes, it had the opposite effect to 9-ODA, which induced the firing of ORNs. We speculate that HOB acts as an inverse agonist to modulate mating behavior at the ORNs level.

The *A. cerana AcerOr11* gene is expressed in neurons specifically tuned to 9-ODA and HOB. However, how those two opposite ligands affect the *AcerOr11* receptor is unknown. Therefore, we cloned and expressed *AcerOr11* in the *Xenopus* oocyte system, which exhibited robust responses to 9-ODA while HOB evoked a reverse, concentration-dependent current. These results again indicated that HOB acts as an inverse agonist of *AcerOr11*. Inverse agonists have been identified in various receptor-ligand interactions, including GABAA, melanocortin receptors, mu-opioid receptors, adrenoceptors, and histamine receptors[24–27]. Currently, the concept of inverse agonists in insect ORs is poorly understood and much less studied. Some experimental evidence suggests that insect ORs have specific inverse agonists. For example, in *C. quinquefasciatus*, OR32 produces regular currents when stimulated with methyl salicylate, while eucalyptol elicits inverse currents, suggesting that eucalyptol might be an inverse agonist[23]. In *Aedes aegypti*, *AaegOR8* was found to be sensitively tuned to (R)-1-octen-3-ol while the structurally unrelated odorant indole inhibited octenol-activated *OR8* and repelled mosquitoes[28]. The potential mechanism of inverse currents can be speculated in three stages: when the receptor is challenged, the agonist interacts with the ligand binding site, triggering an inward current, while the antagonist blocks the binding site to prevent the agonist binding while maintaining spontaneous activity, and inverse agonist "snare" receptor to inactiveness. Our results suggest that HOB to *AcerOr11* generates an additional signal coding, which may execute more complex regulation instead of a simple "on-off" function.

Agonist-inverse agonist combinations may elicit opposite physiological effects. For example, the mushroom psychoactive compound muscimol induces a relaxing effect by activating the GABAA receptor. On the other hand, beta-carbolines act as inverse agonists and cause convulsive or anxiogenic effects[24]. Similarly, 9-ODA promotes mating behavior, while HOB inhibits mating. In summary, our results suggest that mating in

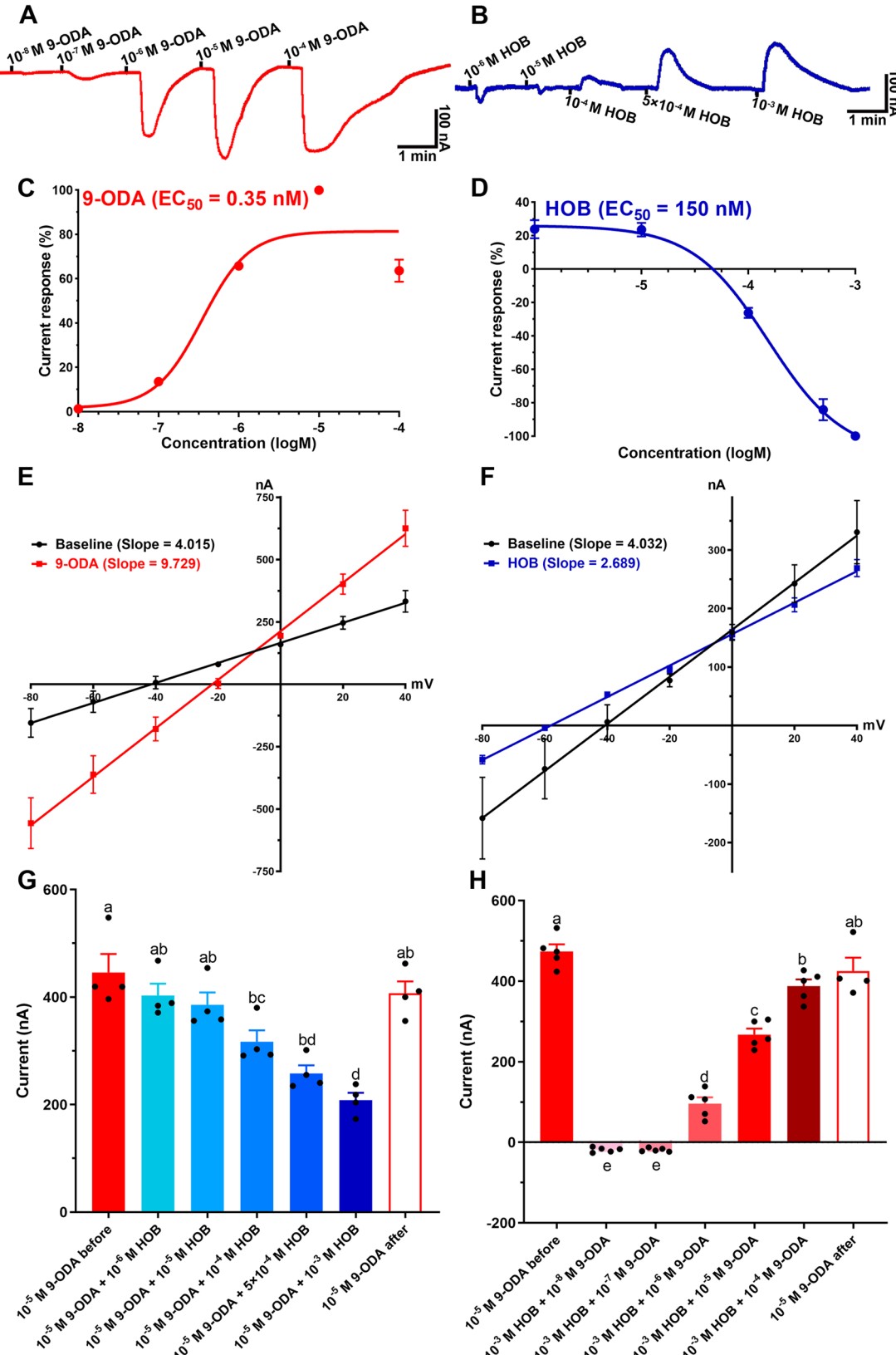

**Fig. 6 | TEVC response of AcerOr11 to 9-ODA and HOB.** Representative trace of currents recorded from AcerOr11/AcerOrco-expressing oocytes when challenged with increasing doses of **A** 9-ODA from $10^{-8}$ M to $10^{-4}$ M and **B** HOB from $10^{-6}$ M to $10^{-3}$ M. Dose-response curves for **C** 9-ODA activation of AcerOr11/AcerOrco (EC$_{50}$ = 0.35 nM, mean ± SEM, $N$ = 5–6) and **D** HOB activation of AcerOr11/

AcerOrco (EC$_{50}$ = 0.15 μM, mean ± SEM, $N$ = 5–7). **E, F** I–V curves of 9-ODA (red) and HOB (blue) at $10^{-5}$ M and $10^{-3}$ M in the voltage range of −80 to +40 mV ($N$ = 6). **G, H** HOB elicited dose-dependent inhibition of 9-ODA-induced responses of AcerOr11/AcerOrco-expressing oocytes (mean ± SEM, $N$ = 4–5, $p < 0.05$, One-way ANOVA followed by Turkey's test).

honeybees is modulated by the two key QMP components, 9-ODA and HOB. *AcerOr11* is not merely an on/off switch but rather functions as a molecular olfactory dimmer.

## Methods

### Honeybees

Honeybees (*A. cerana*) used in this study were provided by the Jilin Provincial Institute of Apicultural Sciences (JLAS), China. The bee colony was originally collected from Dunhua, China (43° 51′ 46″ N, 128° 20′ 30″ E) and has been reared since 2020 in the conservation area of JLAS in a natural environment. Before beginning the experiments, the honeybees were cultured in an artificial incubator (Boxun, China) at 30 °C, 70% humidity, with a 16-h photoperiod. Adult bees were fed with a 10% sucrose solution.

### Gas chromatographic-mass spectrometry (GC-MS) analysis

Heads from the virgin queens, mated queens (mated on day 6 or 7), drones, and workers were collected from 12- to 15-day-old honeybees. The compounds of the mandibular gland were extracted by placing the heads in 200 μL dichloromethane for at least 24 h. The extracts were then dried under a stream of nitrogen, and the remainders were dissolved in 20 μL internal standard solution (octanoic acid and tetradecane in dichloromethane) and 20 μL N,O-Bis (trimethylsilyl) trifluoroacetamide. The mandibular gland pheromone mixes were separated by a GC-MS system (Thermo Fisher Scientific, USA), which was equipped with an HP-INNOWax capillary column, in the split-less mode on a methyl silicone-coated fused silica column (HP - 1MS, 25 m × 0.20 mm × 0.33 μm). Helium gas was used as a carrier gas at a constant flow rate of 1 mL/min. The oven temperature was set to 100 °C for 2 min and then increased to 250 °C at a rate of 10 °C per minute. The final temperature was maintained for 10 min. The compounds were identified by comparing their retention times and mass fragmentation with the known reference compounds. 9-ODA and HOB were further quantified by injecting corresponding standard compounds.

### Behavioral assay

To explore the biological effect of 9-ODA and HOB on drones, we designed a binary-choice Y-tube olfactometer assay. Briefly, 10 μL of a test compound solution was applied to a 25 × 15 mm filter paper and then placed in one arm of the olfactometer (15 cm base, 10 cm arm length, and 2 cm diameter) as the odorant source. The solvent in the other arm was the mock control. Bee responses within 5 min were scored as "made a choice" when an individual moved at least 2/3 into one arm. More than thirty honeybees were used in each behavioral assay for a series of test compound concentrations.

### Electroantennography recording

In the EAG assay, the tip of the honeybee antennae was cut and covered with a conductive gel (Parker Laboratories Inc., USA), and the honeybee head was attached to the reference electrode. In a preliminary test, we found that both 9-ODA and HOB are difficult to dissolve in hexane. Thus, we first dissolved them in ethanol and then diluted them with paraffin oil to the desired dose. Ethanol alone, diluted with paraffin oil, was used as a negative control. A 10 μL stimulus was loaded onto a 5.0 × 0.5 cm filter paper strip and then inserted into a syringe with a continuous flow of 500 mL/min and an air humidity of 60–70%. The pulse flow duration was 0.2 s, and the antenna response was recorded for 5 s. To ensure EAG sensitivity restoration, we had 1-min gaps between two stimulations. For the EAG inhibition test, 9-ODA and HOB were first separated, and then the two pulse flows were mixed at the end of the tube before puffing against the *A. cerana* antennae. The negative control was performed both at the beginning and the end of each preparation. The EAG data were normalized using the negative control data.

### Single sensillum recording

In the SSR test, 12- to 15-day-old drones were wedged into a 1 mL plastic pipette tip, and the protruding head was fixed to the rim of the pipette tip with dental wax. One of the exposed antennae was stuck to a coverslip with

double-sided tape under a microscope (LEICA Z16 APO, Germany). The reference tungsten electrode was inserted into the eye, and spikes were recorded by inserting the tungsten electrode into the base of a sensillum until a stable electrical signal with a high signal-to-noise ratio was achieved. For stimulus delivery, 10 μL of the QMP component was added on a 1 cm × 2.5 cm filter paper strip and then inserted into a Pasteur pipette. A flow of purified and humidified air (2 L/min) was continuously maintained on the antennae through a 14-cm-long metal tube controlled (Syntech Hilversum, Netherlands) by a Syntech stimulus controller (CS-55 model, Syntech, Germany). The two antennae were exposed to a stimulus for 500 ms with airflow of 0.6 L/min through a Pasteur pipette. The action potential signals were amplified using a pre-amplifier (IDAC-4 USB System, Syntech, Germany) and visualized by the Autospike 32 software (Syntech, Germany). The number of induced spikes were calculated as the subtraction from the firing spike number by spontaneous spikes number before the stimulus.

### In situ hybridization

Antisense and sense digoxigenin- and biotin-labeled riboprobes of *AcerOr11* were synthesized using linearized pGEMHE plasmids containing appropriate insertion sequences as a template using the DIG and Biotin RNA Labeling Mix (Roche, Germany) and T7 RNA Polymerase (Roche, Germany). Subsequently, the probe was digested into approximately 400 base fragments by incubating in carbonate buffer (80 mM NaHCO$_3$, 120 mM Na$_2$CO$_3$, pH 10.2).

Antennae of 12- to 15-day-old honeybees of three castes were collected and then embedded in a Tissue-Tek optimal cutting temperature compound (Sakura Finetek, USA). Longitudinal and transverse sections (10 μm thick) through antennae were prepared using the Cryostar NX50 cryostat (ThermoFisher, USA) at −25 °C. The sections were thaw-mounted on adhesive microscope slides (Citotest, China) and immediately utilized for in situ hybridization experiments.

Tissues were fixed in 4% paraformaldehyde, and slides were washed with phosphate-buffered saline (PBS) buffer and 0.6% HCl respectively. For pre-hybridization, slides were immersed in 50% formamide with 2× saline-sodium citrate (SSC) for 1 h at 60 °C. Afterward, the slides were added with 100 μL of the hybridization buffer containing the labeled probe for *AcerOr11* and incubated at 60 °C for a minimum of 16 h. After hybridization, slides were washed in 0.2× SSC, followed by treatment with a 1% blocking solution (Roche, Germany) prepared in tris-buffered saline (TBS) buffer with 0.03% Triton X-100. Anti-Digoxigenin-AP, Fab fragments (catalog number 11093274910, Roche, Germany) and NBT/BCIP (Roche, Germany) were used to detect the DIG-labeled probe under an Upright Microscope BX51 (Olympus, Japan). Anti-Digoxigenin-Fluorescein, Fab fragments (catalog number 11207741910, Roche, Germany) were used to detect the biotin-labeled probe, and the fluorescence signals were visualized under a Zeiss LSM 880 confocal microscope (Zeiss, Jena, Germany) using excitation at 550 nm.

### RNA extraction, gene cloning, and quantitative PCR

RNA samples from different tissues, including the chemosensory organs (antenna, proboscis) and non-chemosensory body parts (thorax, abdomen, and legs), were collected from 15 bees per caste. Total RNA was extracted using the TRIzol reagent (Invitrogen, USA) following the manufacturer's protocols. The concentration and purity of the extracted RNA were measured by a NanoDrop 2000 spectrophotometer (Thermo Fisher Scientific, USA) and 1% agarose gel electrophoresis, respectively.

For PCRs, we used gene-specific primers for *AcerOrco* and *AcerOr11* (Supplementary Table 2). First-strand cDNA synthesis was performed using the TransScript One-Step gDNA Removal and cDNA Synthesis SuperMix (Transgen Biotech, Beijing, China). PCR was performed with the TSINGKE TSE101 PCR enzyme mix (TsingKe Biotech, Beijing, China) at the following conditions: 2 min at 98 °C; followed by 35 cycles of 98 °C for 10 s, 50–60 °C for 10 s, and 72 °C for 20 s; and final extension for 5 min at 72 °C. PCR-amplified products were examined and gel purified using the SanPrep Column DNA Gel Extraction Kit (Sangon Bio, Shanghai, China).

The purified PCR products were subcloned into a pGEMHE vector between the BamHI and HindIII restriction sites using the pEASY-Uni Seamless Cloning and Assembly Kit (Transgen Biotech, Beijing, China).

For the qPCR assay, the cDNA sample was quantified using 1 μg of total RNA, and *β-actin* (GenBank accession: HM640276.1) was used as an internal control gene. Primers for *AcerOr11* and *AcerOrco* were designed using Primer 3 (Supplementary Table 2). RT-qPCR was conducted on a LightCycler 480 II Detection System (Roche, Switzerland) with TransStar Tip Top Green qPCR Supermix (Transgen Biotech, China) at the following conditions: 94 °C for 30 s, followed by 45 cycles of 94 °C for 5 s, 55 °C for 15 s, and 72 °C for 10 s. qPCR data were analyzed by the $2^{-\Delta\Delta CT}$ method.

## Deorphanization of AcerORs in the *Xenopus* oocyte system

The cRNAs with the templates, the linearized pGEMHE vector containing of *AcerOrco* and *AcerOr11*, using the mMESSAGE mMACHINE T7 Kit (Ambion, USA) following the manufacturer's instructions. The cRNAs were adjusted concentration of 200 ng/μL in nuclease-free water and 18.4 nL of *AcerOr11* with same amount of *AcerOrco* cRNAs were microinjected into *Xenopus laevis* oocytes at vegetal pole in stages V or VI using a NanoLiter 2000 injector (World Precision Instruments, Sarasota, USA). Subsequently, oocytes were incubated at 18 °C for 2–8 days in Barth's solution (96 mM NaCl, 2 mM KCl, 5 mM MgCl$_2$, 0.8 mM CaCl$_2$, and 5 mM HEPES; pH 7.6) supplemented with 50 μg/mL tetracycline, 100 μg/mL streptomycin, and 500 μg/mL sodium pyruvate.

A two-electrode voltage-clamp (TEVC) technique was used to record the ion channel-induced currents in *Xenopus* oocytes at a holding potential of −80 mV. For I–V curves, the holding potentials were held between −80 and +40 mV. Signals were amplified with an Axonclamp 900 A amplifier (Molecular Devices, San Jose, USA). Data acquisition and analysis were performed using Axon Digidata 1550B and pCLAMP10 software, using 50 Hz low-pass filters and digitization at 1 kHz (Molecular Devices, USA). The stock solutions (1 M) of all compounds were prepared in DMSO and then diluted with Ringer buffer. Data collected in TEVC were analyzed by Clampfit 10 software. *AcerOr11* expressed ORs were deorphanized against a panel of 163 odorants, including honeybee pheromones and plant volatiles (Supplementary Table 3). The I–V curves were measured by applying a series of voltages, −80, −60, −40, −20, 0, +20, and +40 mV to the tested eggs and current changes were observed.

## Statistics and reproducibility

The significant difference of behavioral assay and SSR were analyzed by using T-test (two-tailed), Wilcoxon signed-ranked test was used for dose-dependent curve in EAG, the inhibitory effect of HOB in TEVC and EAG were analyzed by one-way ANOVA followed by Tukey's multiple comparison test ($p < 0.05$) after checking the normality and homogeneity of variance. All the statistic were performed by SPSS v25.0 (IBM) and visualized by GraphPad Prism v8.0 (GraphPad Software).

## Reporting summary

Further information on research design is available in the Nature Portfolio Reporting Summary linked to this article.

## Data availability

All relevant data linked to manuscript are available in Supplementary materials and the raw data are available from corresponding author on reasonable request.

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

## Acknowledgements

This work was supported by the National Key Research and Development Program (2023YFE0113600), Israel Science Foundation grants (No. 719/21). We thank Dr. Pingxi Xu (Department of MCB, UC Davis) for critically reading an earlier manuscript draft; Dr. Yuanhong Wang (School of Chemistry, Northeast Normal University) for the help with GC-MS analysis; Xiuli Wang and Fan Zhang (School of Life Sciences, Northeast Normal Univerisity) for assistance in in situ hybridization; Dr. Baiwei Ma (Chinese Academy of Agricultural Sciences), Dr. Shuai Liu (Department of Plant Protection, Jilin Agricultural University) and Cuiwei Liu for the guidance in SSR experiments.

## Author contributions

Y.W. and J.D.B. designed this study; H.K., S.D., and X.M. performed this research; C.X. provided and reared the honeybee colony; Y.W. and H.K. analyzed the data; Y.W., J.D.B., and B.R. wrote and revised manuscript.

## Competing interests

The authors declare no competing interests.
