## [Peer review file · Communications Biology]

[Reviewer 1 major comments]:

Reviewer #1 (Remarks to the Author):

The manuscript by Haoqin Ke et al., entitled “The dual coding of a single sex pheromone receptor in regulating the mating behavior of Asian Honeybee *Apis cerana*” represents a truly interesting and sound contribution to the understanding of chemical communication of honeybees, more specifically the Asian species *Apis cerana*.

It builds on the amassed body of knowledge on the European *A. mellifera* in terms of the queen mandibular pheromone (QMP), its individual components and their biological role, including the identification of the olfactory receptor AmelOR11, responsible for the detection of the main QMP component 9-oxo-(E)-2-decenoic acid (ODA). The present manuscript extends the interest to the congeneric honeybee *A. cerana* and its potential ODA receptor AcerOR11, orthologous to AmelOR11. The study aims to confirm its putative function as ODA receptor and also further explores the olfactory function of another QMP component with hitherto poorly understood biological significance, methyl p-hydroxybenzoate (HOB).

The paper brings new and important evidence in two aspects. First, it convincingly confirms the role of ODA as attractant of drones, analogously to *A. mellifera*, and the role of AcerOR11 as ODA receptor in *A. cerana*. To this goal, a range of methods is implemented, starting from attraction choice bioassays, through EAG and SSR recordings, situating the ODA detection into placoid sensilla (just like in *A. mellifera*), to in situ hybridization and heterologous expression of AcerOR11 in the *Xenopus* system providing proof of OR11 to be the responsible receptor.

And second, supported by GC MS analysis showing the absence of HOB in virgin queens while its expected presence in mated queens, the paper explores the potential role of HOB as a semiochemical responsible for loss of attractiveness to drones of the queens that are mated. Once again, a series of experiments including behavioral, EAG, SSR and patch clamp on *Xenopus* oocytes is used. The paper shows that HOB acts as an antagonist mitigating the drone attraction effect of ODA. The study further shows that HOB directly acts through the ODA receptor AcerOR11, and thus may be classified as an inverse agonist. This finding may be particularly important since the role of HOB as a secondary component of the multicomponent QMP has not yet been fully clarified in any *Apis* species, including *A. mellifera*.

By its findings, the manuscript represents a nice contribution to the field and I would like to see it published in *Commun Biol*. Yet, the manuscript also suffers important flaws that need to be addressed prior to publication. I have two major remarks, listed below:

[Response]: Thank you for comments on our manuscript. We have addressed your concerns in revised manuscript, paying particular attention to enhancing the English language and grammar accuracy

throughout the revised manuscript. Our detailed responses are as following:

1. The paper does not address sufficiently the body of knowledge on HOB acquired in *A. mellifera* and needs to provide a broader context in this respect. For instance, it does not mention the previous papers proposing its role, including those that quantified HOB in queens of different mating statuses and found HOB to be present in the QMP in virgin queens (e.g. doi.org/10.1007/s00265-008-0581-9 and others). The novelty of HOB being absent in virgins needs a proper discussion in context of papers with contrasting results.

[Response]: Thank you for your valuable suggestions. To enrich the knowledge of HOB, we add more contents in discussion part about HOB in revised manuscript, after checking their results, we found it is truly had some HOB in virgin queen's QMP extracts, but according to their results, the mated queens obviously had a higher concentration of HOB compared with virgin queens, which is consist with our findings, we still do not know why we didn't detect it in our virgin queens, maybe it is due to the sampling time, and they did not mention the exact sampling time of the QMP they used, but all the other studies are not detect HOB in virgin queens, the detailed added discussion part were as follows:

Line 326-329: HOB, a pheromone signal only released by mated queens, exerts a "stop mating" function on drones. While a few studies have also reported the presence of tiny levels of HOB in virgin queens¹³, most of the QMP studies, as well as ours, showed HOB is fully absent in virgin queens^{4,14}.

4. Plettner, E. *et al.* Species- and caste-determined mandibular gland signals in honeybees (*Apis*). *J. Chem. Ecol.* **23**, 363-377, doi:10.1023/B:JOEC.0000006365.20996.a2 (1997).

13. Strauss, K. *et al.* The role of the queen mandibular gland pheromone in honeybees (*Apis mellifera*): honest signal or suppressive agent? *Behav. Ecol. Sociobiol.* **62**, 1523-1531, doi:10.1007/s00265-008-0581-9 (2008).

14. Villar, G., Hefetz, A. & Grozinger, C. M. Evaluating the Effect of Honey Bee (*Apis mellifera*) Queen Reproductive State on Pheromone-Mediated Interactions with Male Drone Bees. *J. Chem. Ecol.* **45**, 588-597, doi:10.1007/s10886-019-01086-0 (2019).

2. The manuscript suffers with large numbers of problems in grammar and syntax, often leading to sentences that lose sense. There is big room for improvement in this respect since in the current version these errors truly put in risk the scientific clarity. I attach to this review an annotated pdf of the submission to highlight some of these problematic points directly in the text.

In the annotated version the authors may find also a few other comments to the scientific aspects.

[Response]: Thank you for your carefully check. We carefully review the annotated PDF and make the corresponding corrections point by point, we appreciate your annotated PDF in improving the language quality, the detailed revision as follows:

Line16: which strategy?

[Response]: Actually, “this reproductive strategy” means virgin queens usually mates only once before founding a colony. To avoid the misunderstanding, we add more sentences in addressing this in abstracts, the sentences were listed here:

Line 15-16: In Asian honeybees, virgin queens typically mate only once before founding a colony. This behavior is controlled by the queen-released mandibular pheromone (QMP).

Line20: Give a citation that it acts so...

[Response]: Previously, less studies focusing on the stop mating mechanism in Asian honeybees, but it is mainly our findings, to avoid confusion, we changed this sentence into “However, how the queens prevent additional mating remains elusive.” Thank you for the reminds.

Line29: the SSR and EAG experiments did not show that the sensilla narrowly respond to HOB.

At this stage of knowledge, HOB seemed to disrupt ODA detection which does not mean to narrowly respond.

[Response]: Indeed, in EAG assays, drone didn’t show obvious inverse response to HOB, but in SSR, this phenomenon is significant, and the reason for it is not obvious in SSR figures may be due to the low frequency of the spontaneous activity in neuron A naturally, even neuron A’s activity is fully suppressed, the decrease of the spikes in SSR is not obvious in figure. However, the statistic results showed HOB could significantly ($P < 0.001$) reduced neuron A’s spontaneous frequency. And the other evidence point to the fact that HOB and 9-ODA were both functional on the same receptor and same neuron, rather than just block the 9-ODA’s response, which might couldn’t be simply defined as a “antagonist”.

Line30: italics

[Response]: Thank you for your carefully check, the word is corrected.

Line 24-26: Deorphanization of *AcerOr11* in *Xenopus* oocyte system showed 9-ODA induces robust inward (regular) currents, while HOB induces inverse currents in a dose-dependent manner.

Line38-48: This entire paragraph is pretty much redundant and does not bring any information needed for further understanding. There is no need to say that population dynamics in insects are influenced by some factors, and there is no need to speak about ocimene and other semiochemicals, etc. The manuscript can easily start with the second paragraph targeting directly the topic of the study.

What “vegetative diversity on diets” means?

[Response]: That is really a good advice. In the revised manuscript, we have removed the first paragraph about population dynamics in introduction and discussion parts, to make the manuscript more focus.

Line50-54: These two sentences show essentially the same and one sentence with non-redundant information would do the same job.

Moreover, repeated linking of chemical signaling of mating status with the impact on population dynamics is in my view erroneous and occurs on multiple places of the manuscript. The queen should have enough sperm to make the colony functional after the mating flight. To avoid further mating is of course important for the function of the colony but does not directly translate into population dynamics.

[Response]: Thank you for your guidance. We removed the population dynamics contents in this part and throughout the manuscript, to be more rigorous, we less discuss the biological meaning of this “stop mating” behavior in queens is responsible for saving energy for other important tasks such as oviposition.

Line 37-42: A Virgin honeybee queen usually mates only once with several drones, and for the rest of her prolific life, she will not engage in subsequent mating events ^{1,2}. The queen mandibular pheromone (QMP) plays a key role in regulating colony reproduction. QMP directly triggers the mating behavior and provides information about the mating status of queens. It also inhibits the development of worker ovaries by changing relevant gene expression ³.

1. Butler, C. G. The mating behavior of the honeybee (*Apis mellifera* L.). *J. Entomol.* 46, 1-11, doi:10.1111/j.1365-3032.1971.tb00103.x (2009).
2. Gary, N. E. & Marston, J. Mating behavior of drone honey bees with queen models (*Apis mellifera* L.). *Anim. Behav.* 19, 299-304, doi:10.1016/s0003-3472(71)80010-6 (1971).
3. Sandoz, J. C., Deisig, N., de Brito Sanchez, M. G. & Giurfa, M. Understanding the logics of pheromone processing in the honeybee brain: from labeled-lines to across-fiber patterns. *Front. Behav. Neurosci.* 1, 5, doi:10.3389/neuro.08.005.2007 (2007).

Line59: please, provide reference.

[Response]: Thank you for your advice, we added two more references to prove HVA is absent in *A. cerana*.

Line 46-47: In *A. cerana*, 9-ODA, 9-HDA and HOB are also the dominant component of QMP, while HVA is absent ^{4,5}.

4. Plettner, E. *et al.* Species- and caste-determined mandibular gland signals in honeybees (*Apis*). *J. Chem. Ecol.* **23**, 363-377, doi:10.1023/B:JOEC.0000006365.20996.a2 (1997).
5. Keeling, C. I., Otis, G. W., Hadisoelilo, S. & Slessor, K. N. Mandibular gland component analysis

in the head extracts of *Apis cerana* and *Apis nigrocincta*. *Apidologie* **32**, 243-252, doi:10.1051/apido:2001126 (2001).

Line63: please, provide reference

[Response]: We have added more references to support this idea.

Line 50-51: However, the role of HOB remains poorly understood, as single secondary QMP components are not attractive to drones ^{6,7}.

6. Pankiw, T. *et al.* Mandibular gland components of european and africanized honey bee queens (*Apis mellifera* L.). *J. Chem. Ecol.* **22**, 605-615, doi:10.1007/bf02033573 (1996).

7. Keeling, C. I., Slessor, K. N., Higo, H. A. & Winston, M. L. New components of the honey bee (*Apis mellifera* L.) queen retinue pheromone. *Proc. Natl. Acad. Sci. USA* **100**, 4486-4491, doi:10.1073/pnas.0836984100 (2003).

Line68: please, provide references

[Response]: We have added more references to support this idea.

Line 55-56: which expresses the *A. mellifera* odorant receptor 11 gene (*AmelOr11*) ⁸.

8. Kaissling, K. E. & Renner, M. Antennale Rezeptoren für Queen Substance und Sterzelduft bei der Honigbiene. *Zeitschrift für Vergleichende Physiologie* **59**, 357-361, doi:10.1007/bf00365967 (1968).

Line75: QMP composition is not olfactory system

[Response]: Thank you for your careful check. The sentence was corrected into “they differ in their olfactory systems, as well as QMP composition”

Line 61-64: Although *A. mellifera* and *A. cerana* share overall similar morphology and social behavior, they differ in their olfactory systems, as well as QMP composition, the number of odorant receptors (Ors), and antennal lobe topology ¹¹.

11. McKenzie, S. K., Fetter-Pruneda, I., Ruta, V. & Kronauer, D. J. Transcriptomics and neuroanatomy of the clonal raider ant implicate an expanded clade of odorant receptors in chemical communication. *Proc. Natl. Acad. Sci. USA* **113**, 14091-14096, doi:10.1073/pnas.1610800113 (2016).

Line77-80: AcerOR11 is targeted here for some reason, and the reason is that it is orthologous to AmelOR11. This should be made clear here.

[Response]: Thank you for pointing out this problem. We add one sentence here as “Our transcriptome data showed the 9-ODA receptor orthologous is also existed in *A. cerana*, which shed a light on discovery the olfactory pathway of sex pheromone sensing in Asian honey bees.”

Line 65-70: In *A. cerana*, odorant binding protein 11 (AcerOBP11) demonstrates strong binding

affinities for both 9-ODA and HOB¹². However, the specific ORs responsible for detecting 9-ODA and HOB in *A. cerana* remain unclear. Our transcriptome data showed that the 9-ODA receptor ortholog also exists in *A. cerana*, which shed a light on discovery the olfactory pathway on sex pheromone sensing in Asian honey bees.

12. Song, X. M. *et al.* Various Bee Pheromones Binding Affinity, Exclusive Chemosensillar Localization, and Key Amino Acid Sites Reveal the Distinctive Characteristics of Odorant-Binding Protein 11 in the Eastern Honey Bee, *Apis cerana*. *Front. Physiol.* **9**, 422, doi:10.3389/fphys.2018.00422 (2018).

Line81: This should be put into context of the previous findings. Is it new? Or is it notorious? What about other *Apis* species and HOB and mating status?

[Response]: Thank you for the suggestions. The paper you listed truly showed low levels of HOB is detected in virgin queens, however, in most other QMP studies, as well as ours, showing HOB is fully absent in virgin queens. To avoid misunderstand, we corrected this sentence into “In this study, we found that HOB is only released by mated queens, which is consist with most QMP studies [X], and could significantly reduce the attraction of drones to 9-ODA.

Line 71-72: In this study, we found that HOB, released only by mated queens, significantly reduces the attraction of drones to 9-ODA.

Line83: electroantennography

[Response]: Thank you for the check, corrected throughout the manuscript.

Line 72-74: his inverse effect of HOB was further validated by *in vivo* electrophysiological assays, electroantennography (EAG), and single sensillum recording (SSR).

Line88-89: redundant.

[Response]: It was deleted.

Line 77-78: This study aimed to explore how QMP regulates reproduction at the olfactory sensing level of *A. cerana*.

Line105: two columns are listed. What was the separation column?

[Response]: Thank you for your reminder. BSTFA is the short name for the “N,O-Bis (trimethylsilyl) trifluoroacetamide”, to avoid any misunderstands, the “(BSTFA)” was removed.

Line 92-94: The extracts were then dried under a stream of nitrogen, and the remainders were dissolved in 20 μ L internal standard solution (octanoic acid and tetradecane in dichloromethane) and 20 μ L N,O-

Bis (trimethylsilyl) trifluoroacetamide.

Line110: to

[Response]: Corrected.

Line 99-100: Helium gas was used as a carrier gas at a constant flow rate of 1 mL/min. The oven temperature was set to 100 °C for 2 min and then increased to 250 °C at a rate of 10 °C per minute.

Line113: I assume that using GCMS also allowed to identify the compounds based on their mass fragmentation.

[Response]: Thank you for your carefully check, indeed, both retention times and fragmentation patterns were used in identifying target compounds, the word “mass fragmentation” was added to the sentence.

Line 100-102: The final temperature was maintained for 10 min. The compounds were identified by comparing their retention times and mass fragmentation with the known reference compounds.

Line117: please, add “solution” or “dilution”

[Response]: Thank you for the check, the word “solution” was added.

Line 106-108: Briefly, 10 µL of a test compound solution was applied to a 25 × 15 mm filter paper and then placed in one arm of the olfactometer (15 cm base, 10 cm arm length, and 2 cm diameter) as the odorant source.

Line122: plural

[Response]: “concentration” was corrected into “concentrations” in revised version.

Line 110-112: More than thirty honeybees were used in each behavioral assay for a series of test compound concentrations.

Line124: both ends of the antenna? Or ends of both antennae? Please clarify here.

[Response]: The word was corrected into “The tip of the antennae”.

Line 114-116: In the EAG assay, the tip of the honeybee antennae were cut and covered with a conductive gel (Parker Laboratories Inc., USA), and the honeybee head was attached to the reference electrode.

Line149: plural

[Response]: Corrected.

Line 139-140: The two antennae were exposed to a stimulus for 500 ms with airflow of 0.6 L/min

through a Pasteur pipette.

Line153-154: rather “subtraction of the spontaneous spikes FROM the firing spike numbers.”

[Response]: “of” was corrected into “from” instead.

Line 142-144: The number of induced spikes were calculated as the subtraction from the firing spike number by spontaneous spikes number before the stimulus.

Line162: which caste?

[Response]: Sorry for misleading, three castes were all included, we add more details here.

Line 152-154: Antennae of 12- to 15-day-old honeybees of three castes were collected and then embedded in a Tissue-Tek optimal cutting temperature compound (Sakura Finetek, USA).

Line224: analyzed

[Response]: Corrected

Line 214-215: Data collected in TEVC were analyzed by Clampfit 10 software.

Line231: contents

[Response]: Corrected

Line 221-222: To measure the contents of QMPs in *A. cerana*, we performed GC-MS analysis on 12- to 15-day-old virgin queens, mated queens as well as drones and workers.

Line233: plural

[Response]: Corrected

Line 222-224: 9-ODA and 9-HDA, two main components in QMPs, were detectable in both virgin and mated queens (Figure 1A, B, S1).

Line234: “mated queens other than virgin”? Please, reword.

[Response]: “HOB was only detected in mated queens other than virgin” was corrected into “HOB was only detected in the mated queens and not in virgin queens”.

Line 224-225: However, interestingly, HOB was only detected in the mated queens and not in virgin queens (Figure 1B).

Line235: please reword

[Response]: Changed into “inspired us”.

Line 225-226: This inspired us that HOB might exercise some functions on the post-mating regulation.

Line238: past tense

[Response]: “aim” was changed into “aimed”

Line 228-229: Thus, we aimed to test the behavioral effects of HOB on drones, using the Y-tube olfactometer assay.

Line238: “drones. We” Please, split in two sentences or reword.

[Response]: The two sentences were reworded.

Line 228-229: Thus, we aimed to test the behavioral effects of HOB on drones, using the Y-tube olfactometer assay.

Line240: please remove “were” or exchange “at” for “if”

[Response]: Corrected.

Line 229-231: First, we used 9-ODA as a stimulus and found that it has a significant attractive effect on drones only at the highest concentration (100 μg) ($p < 0.05$, two-tailed, T-test).

Line244: doses

[Response]: Corrected.

Line 234-235: When we mixed 100 μg of 9-ODA with different concentrations of HOB (0.1-100 μg)

Line245: was

[Response]: Corrected

Line 235-236: we found that 9-ODA’s attraction was suppressed by HOB at concentration beyond 1 μg .

Line256: “in any caste”

[Response]: We gratefully appreciate for your valuable suggestion. We have corrected in revised manuscript.

Line 246-247: On the contrary, HOB did not elicit any significant antennal response in any castes (Figure 3B and S2).

Line262: replace with “and”

[Response]: Thank for your careful checks. We have replaced it.

Line 253-254: and the effects were significantly lower than those from 9-ODA alone (Figure 3D).

Line266: Please, delete and start the sentence as “Three types...”

[Response]: We sincerely thank you for your careful reading. We have corrected it.

Line 258-260: Three types of chemosensory sensilla were observed in the *A. cerana* drone’s antenna, including the sensilla trichodea, placodea, and basiconica.

Line277: neither...nor...changed

[Response]: Thank you for your nice suggestion. We will pay more attention.

Line 269-270: Intriguingly, neither 9-ODA nor HOB changed the spontaneous activity of B and C neuron (Figure 4D).

Line283: implicating that

[Response]: We are very sorry for our negligence, and we have corrected in revised manuscript.

Line 275-276: Overall, these results implicating that HOB could act as an inverse agonist at the ORNs level.

Line285: quantifying AcerOR11 in different tissues does not help understanding “the potential molecular mechanism”.

[Response]: This sentence was corrected into “To check the expression profile of *AcerOr11*.”

Line 278: To check the expression profile of *AcerOr11* underlying the physiological response...

Line288: So, is it expressed only in the drone antenna or little expressions were also detected elsewhere?

[Response]: The word “only” was corrected into “abundantly”

Line 280-282: We found that *AcerOr11* is abundantly expressed in drone antennae, while little expression was also detected in the queen and worker antennae (Figure S4).

Line289: This is a very good hint the AcerOR11 is important but not a proof yet. The proof (and very convincing proof) will only come later in this chapter

[Response]: This sentence was moved to the end of this chapter.

Line293: “bees from the castes”? Please, reword.

[Response]: Corrected.

Line 284-285: The DIG-labeled riboprobes for *AcerOr11* were applied to transversal antennal sections of bees from three castes.

Line295: does not make much sense, please, reword.

[Response]: The sentence was corrected.

Line 287-289: Only a few cells in workers' antennae expressed *AcerOr11*, while *AcerOr11* was fully undetectable in queens' antennae (Figure 5C, D, L, M).

Line298: areas

[Response]: Corrected.

Line 290-291: *AcerOr11*-labeled areas were mainly distributed in dendrite-like structures of ORNs housed in sensilla placodea (Figure 5K).

Line306: singular

[Response]: Corrected.

Line 297-298: We functionally expressed *AcerOr11* in the *Xenopus laevis* system and screened it with a 163-compound panel.

Line310: response of AerOR11 to HOB, not the opposite.

[Response]: It is true as you suggested that it should be the “response of *AcerOr11* to HOB”, and we have corrected in revised manuscript.

Line 302-303: To further confirm the inverse response of *AcerOr11* to HOB, a current-voltage (I-V) curve was constructed using 9-ODA and HOB as stimuli.

Line320: please, reword

[Response]: Reword.

Line 312-313: We fixed the concentration of 9-ODA to 10^{-5} M for a saturated response while increasing the HOB concentration from 10^{-6} M to 10^{-3} M.

Line327: series

[Response]: Corrected.

Line 318-320: Subsequently, we used 9-ODA alone for a series of concentrations tests and found that the response to 9-ODA was recovered (Figure 6G).

Line334: references

[Response]: We have added reference to support this viewpoint. Thank you for your reminder.

Line 326-329: HOB, a pheromone signal only released by mated queens, exerts a “stop mating” function on drones. While a few studies have also reported the presence of tiny levels of HOB in virgin queens 13, most of the QMP studies, as well as ours, showed HOB is fully absent in virgin queens

4,14.

4. Plettner, E. et al. Species- and caste-determined mandibular gland signals in honeybees (*Apis*). *J. Chem. Ecol.* 23, 363-377, doi:10.1023/B:JOEC.0000006365.20996.a2 (1997).

13. Strauss, K. et al. The role of the queen mandibular gland pheromone in honeybees (*Apis mellifera*): honest signal or suppressive agent? *Behav. Ecol. Sociobiol.* 62, 1523-1531, doi:10.1007/s00265-008-0581-9 (2008).

14. Villar, G., Hefetz, A. & Grozinger, C. M. Evaluating the Effect of Honey Bee (*Apis mellifera*) Queen Reproductive State on Pheromone-Mediated Interactions with Male Drone Bees. *J. Chem. Ecol.* 45, 588-597, doi:10.1007/s10886-019-01086-0 (2019).

Line344: Again, I do not see a causal link here to claim this.

[Response]: The sentence was corrected.

Line 338-340: Our behavioral assay results indicate that HOB directly inhibits the mating behavior of drones and might prevent the queens from consuming the energy for redundant mating.

Line349: please, provide a correct name for the compound.

[Response]: Thanks for your careful checks. We have read the reference and correct its name.

Line 342-343: In *Helicoverpa armigera*, female moths release cis-11-hexadecenol (Z11-16: OH) to repel males and avoid non-optimal mating¹⁶.

16. Chang, H. et al. A Pheromone Antagonist Regulates Optimal Mating Time in the Moth *Helicoverpa armigera*. *Curr. Biol.* 27, 1610-1615, doi:10.1016/j.cub.2017.04.035 (2017).

Line352: their.

[Response]: Corrected.

Line 346: With a few exceptions, honeybee queens do not remate in their lifetime.

Line360: please, provide genus name upon first occurrence

[Response]: Thank you for your reminder. We have updated the full name of *Culex quinquefasciatus*.

Line 354-356: In *Culex quinquefasciatus*, eucalyptol significantly decreases the number of spikes in the ab7 sensillum.

Line363: unclear, please reword

[Response]: The sentence was corrected.

Line 356-358: Besides the interreceptor inhibition, here, we show that HOB reduces the activity of 9-ODA at the same ORNs, supporting the hypothesis of the intrareceptor inhibition.

Line367: acts

[Response]: Corrected.

Line 361-362: We speculate that HOB acts as an inverse agonist to modulate mating behavior at the ORNs level.

Line486: here, you can show the retention times of the synthetic standards in a separate chromatogram, or provide some spectra in supplementary.

[Response]: Thank you for your valuable suggestion. As you suggested, we have supplied the retention times of the synthetic standards (9-ODA and HOB) in Figure S1 in “Supplementary materials”.

[Reviewer 3 major comments]:

Reviewer #3 (Remarks to the Author):

This manuscript describes experiments that identify HOB, a substance of the queen mandibular pheromone, as an anti-aphrodisiac in *Apis cerana*. The interesting result is that HOB seems to inhibit the sex pheromone 9-ODA, that usually attracts males, by directly affecting OR11, the odorant receptor tuned to 9-ODA. I found the results very interesting and did not find any experimental or conceptual flaws, although I have never myself worked on odorant perception on the molecular level. What I did not enjoy was the general presentation of the manuscript. Throughout I was wondering what the biological function of the mechanism described here could be. In the discussion the authors then speculate that an anti-aphrodisiac could reduce mating in the colony, which would reduce colony growth. Either I am misunderstanding something or this does not make any sense. The queens do not mate in the colony, and matings do not have any direct effects on egg-laying rates (other than that the queen needs to be mated at least once to lay fertilized eggs). The language requires some editing overall, there are many errors (e.g. mixed plural & singular, inconsistent use of past and present tense, typos), and a few more details on the exact procedures would be nice as well (see below).

[Response]: Thank you for valuable comments on our manuscript. We have paying particular attention to enhancing the English language and grammar accuracy throughout the revised manuscript. Meanwhile, to be more rigorous, we removed the population dynamic introduction and discussion in the revised version, instead, we just mentioned this “stop mating” behavior as an energy saving strategy in queens to be a trade-off to their other important tasks such as oviposition. Our detailed responses were as follows:

line 16 “this strategy” which strategy?

[Response]: Sorry for the confusion. Actually, “this reproductive strategy” means virgin queens usually mates only once before founding a colony. To avoid the confusion, we rewrite this part in abstract, the details are as follows:

Line 15-16: In Asian honeybees, virgin queens typically mate only once before founding a colony. This behavior is controlled by the queen-released mandibular pheromone (QMP).

20 why would one expect that qmp prevents mating?

[Response]: Previously, few studies are focusing on the stop mating mechanism in Asian honeybees, but it is mainly our findings, to avoid confusion, we changed this sentence into “However, how the queens prevent additional mating remains elusive.” Thank you for the reminds.

38 I don't see the connection to population dynamics at all, in particular regarding the environmental factors mentioned in the first sentence

[Response]: Thank you for your professional advice. The reproduction and the maintain of sociality in honeybee colony are intricately linked to chemical signals. However, the regulate of population dynamics are influenced by a variety of factors instead of chemical signals simply. Your suggestion has greatly inspired us. Consequently, in the revised manuscript, we have excised the content pertaining to population dynamics and instead augmented it with more comprehensive discussions on chemical signals between mating behavior, to solidify the foundation for our research.

Is AcerOr11 orthologous to AmelOR11?

[Response]: Thank you for pointing out this problem. We add one sentence here as "Our transcriptome data showed the 9-ODA receptor orthologous is also existed in *A. cerana*, which shed a light on discovery the olfactory pathway of sex pheromone sensing in Asian honey bees."

Line 65-70: In *A. cerana*, odorant binding protein 11 (AcerOBP11) demonstrates strong binding affinities for both 9-ODA and HOB¹². However, the specific ORs responsible for detecting 9-ODA and HOB in *A. cerana* remain unclear. Our transcriptome data showed that the 9-ODA receptor ortholog also exists in *A. cerana*, which shed a light on discovery the olfactory pathway on sex pheromone sensing in Asian honey bees.

12. Song, X. M. *et al.* Various Bee Pheromones Binding Affinity, Exclusive Chemosensillar Localization, and Key Amino Acid Sites Reveal the Distinctive Characteristics of Odorant-Binding Protein 11 in the Eastern Honey Bee, *Apis cerana*. *Front. Physiol.* **9**, 422, doi:10.3389/fphys.2018.00422 (2018).

182 be more specific about the tissues. why this choice? You say including, is this the full list of tissues? (later it becomes clear that tissues were analysed separately to locate the expression of the genes in question)

[Response]: We gratefully appreciate for your valuable comment. In fact, we tested AcerOr11 expression level in chemosensory organs and non-chemosensory body parts (as a negative control). To make it clearer, we rewrite the corelated sentences in revised manuscript.

Line 172-174: RNA samples from different tissues, including the chemosensory organs (antenna, proboscis) and non-chemosensory body parts (thorax, abdomen, and legs), were collected from 15 bees per caste.

231 countents typo

[Response]: Corrected

Line 221-222: To measure the contents of QMPs in *A. cerana*, we performed GC-MS analysis on 12- to 15-day-old virgin queens, mated queens as well as drones and workers.

Fig 1: It would be nice to learn more about sample size etc. here. I find it a bit weird that the data in table S3 indicate that there was never 9-ODA or HOB in workers but in Fig. 1 there are clearly small peaks.

[Response]: Thank you for pointing out this problem in manuscript. In GC-MS analysis, we conducted three biological replications. Although we can see a tiny peak in workers, the contents of it is very small compared to that of queens, it is nearly undistinguishable from the background noise and even hardly detectable in the GC-MS analysis. Meanwhile, in the other two biological replicates, 9-ODA and HOB are fully absent in works. In summary, considering these results, we concluded that 9-ODA and HOB are not released by workers, or to be more rigorous, they were so tiny that even undetectable by GC-MS. Thank you for your suggestions again.

241 “at as”?

[Response]: Corrected.

Line 229-231: First, we used 9-ODA as a stimulus and found that it has a significant attractive effect on drones only at the highest concentration (100 μg) ($p < 0.05$, two-tailed, T-test).

242, 246 I would give separate p-values for the different concentrations in the text

[Response]: Thank you for your comment. We have list p values for different concentrations in revised version.

Line 231-233: At a lower concentration of 0.1-10 μg , the attraction effect of 9-ODA was not significant ($p = 0.5614$ for 0.1 μg ; $p = 0.9580$ for 1 μg ; $p = 0.0668$ for 10 μg) (Figure 2A).

Line 234-237: When we mixed 100 μg of 9-ODA with different concentrations of HOB (0.1-100 μg), we found that 9-ODA's attraction was suppressed by HOB at concentration beyond 1 μg ($p = 0.2302$ for 1 μg ; $p = 0.3771$ for 10 μg ; $p = 0.9414$ for 100 μg) (Figure 2C).

Fig. 4F I don't understand how there can be negative spikes per second. Is this compared to some sort of baseline of the same antenna? When is the baseline measured? Please indicate this in the figure

[Response]: We totally understand your concern. The number of induced spikes were calculated by the firing spike number subtraction from spontaneous spikes number before the stimulus. This information was added in the relevant figure.

301 is it really expressed in the cytoplasm or is the RNA quickly moved out of the nucleus after transcription?

[Response]: Thank you for your rigorous comment. Insect OR is a family belongs to transmembrane receptors families, so the mRNA of ORs should be quickly moved from the nucleus to cytoplasm after transcription.

327 "serious"?

[Response]: Corrected.

Line 318-320: Subsequently, we used 9-ODA alone for a series of concentrations tests and found that the response to 9-ODA was recovered (Figure 6G).

343 "results indicate that HOB directly inhibits the mating behavior of drones and might prevents the bee colony from having an oversized population, trading off the number of offspring for the allocation

of energy resources, including physical resources and parental care.”

I'm not a bee person but that sounds very wrong. The drones do not mate the queen inside the colony. The queen typically takes one mating flight and then lays eggs continuously, independent of further matings. As far as I know this is also true for *Apis cerana*. Now I understand the weird beginning of the introduction talking about colony dynamics. It does not seem to make sense to me.

[Response]: To be more rigorous, we removed the population dynamic discussion in the revised version, instead, we changed this sentence into “results indicate that HOB directly inhibits the mating behavior of drones and might prevents the queens from consuming the energy for redundant mating.”

398 This last paragraph is weird. The findings are interesting in themselves, no need to invoke spurious arguments about conservation. If you invoke conservation, explain how you would use HOB to conserve *A. cerana*.

[Response]: We gratefully thanks for your constructive remarks. As your suggested, we deleted this paragraph in the revised manuscript.

[Reviewer 1 major comments]:

Reviewer #1 (Remarks to the Author):

The manuscript by Haoqin Ke et al., entitled “The dual coding of a single sex pheromone receptor in regulating the mating behavior of Asian Honeybee *Apis cerana*” represents a truly interesting and sound contribution to the understanding of chemical communication of honeybees, more specifically the Asian species *Apis cerana*.

It builds on the amassed body of knowledge on the European *A. mellifera* in terms of the queen mandibular pheromone (QMP), its individual components and their biological role, including the identification of the olfactory receptor AmelOR11, responsible for the detection of the main QMP component 9-oxo-(E)-2-decenoic acid (ODA). The present manuscript extends the interest to the congeneric honeybee *A. cerana* and its potential ODA receptor AcerOR11, orthologous to AmelOR11. The study aims to confirm its putative function as ODA receptor and also further explores the olfactory function of another QMP component with hitherto poorly understood biological significance, methyl p-hydroxybenzoate (HOB).

The paper brings new and important evidence in two aspects. First, it convincingly confirms the role of ODA as attractant of drones, analogously to *A. mellifera*, and the role of AcerOR11 as ODA receptor in *A. cerana*. To this goal, a range of methods is implemented, starting from attraction choice bioassays, through EAG and SSR recordings, situating the ODA detection into placoid sensilla (just like in *A. mellifera*), to in situ hybridization and heterologous expression of AcerOR11 in the *Xenopus* system providing proof of OR11 to be the responsible receptor.

And second, supported by GC MS analysis showing the absence of HOB in virgin queens while its expected presence in mated queens, the paper explores the potential role of HOB as a semiochemical responsible for loss of attractiveness to drones of the queens that are mated. Once again, a series of experiments including behavioral, EAG, SSR and patch clamp on *Xenopus* oocytes is used. The paper shows that HOB acts as an antagonist mitigating the drone attraction effect of ODA. The study further shows that HOB directly acts through the ODA receptor AcerOR11, and thus may be classified as an inverse agonist. This finding may be particularly important since the role of HOB as a secondary component of the multicomponent QMP has not yet been fully clarified in any *Apis* species, including *A. mellifera*.

By its findings, the manuscript represents a nice contribution to the field and I would like to see it published in *Commun Biol*. Yet, the manuscript also suffers important flaws that need to be addressed prior to publication. I have two major remarks, listed below:

[Response]: Thank you for comments on our manuscript. We have addressed your concerns in revised manuscript, paying particular attention to enhancing the English language and grammar accuracy

throughout the revised manuscript. Our detailed responses are as following:

1. The paper does not address sufficiently the body of knowledge on HOB acquired in *A. mellifera* and needs to provide a broader context in this respect. For instance, it does not mention the previous papers proposing its role, including those that quantified HOB in queens of different mating statuses and found HOB to be present in the QMP in virgin queens (e.g. doi.org/10.1007/s00265-008-0581-9 and others). The novelty of HOB being absent in virgins needs a proper discussion in context of papers with contrasting results.

[Response]: Thank you for your valuable suggestions. To enrich the knowledge of HOB, we add more contents in discussion part about HOB in revised manuscript, after checking their results, we found it is truly had some HOB in virgin queen's QMP extracts, but according to their results, the mated queens obviously had a higher concentration of HOB compared with virgin queens, which is consist with our findings, we still do not know why we didn't detect it in our virgin queens, maybe it is due to the sampling time, and they did not mention the exact sampling time of the QMP they used, but all the other studies are not detect HOB in virgin queens, the detailed added discussion part were as follows:

Line 326-329: HOB, a pheromone signal only released by mated queens, exerts a "stop mating" function on drones. While a few studies have also reported the presence of tiny levels of HOB in virgin queens¹³, most of the QMP studies, as well as ours, showed HOB is fully absent in virgin queens^{4,14}.

4. Plettner, E. *et al.* Species- and caste-determined mandibular gland signals in honeybees (*Apis*). *J. Chem. Ecol.* **23**, 363-377, doi:10.1023/B:JOEC.0000006365.20996.a2 (1997).

13. Strauss, K. *et al.* The role of the queen mandibular gland pheromone in honeybees (*Apis mellifera*): honest signal or suppressive agent? *Behav. Ecol. Sociobiol.* **62**, 1523-1531, doi:10.1007/s00265-008-0581-9 (2008).

14. Villar, G., Hefetz, A. & Grozinger, C. M. Evaluating the Effect of Honey Bee (*Apis mellifera*) Queen Reproductive State on Pheromone-Mediated Interactions with Male Drone Bees. *J. Chem. Ecol.* **45**, 588-597, doi:10.1007/s10886-019-01086-0 (2019).

2. The manuscript suffers with large numbers of problems in grammar and syntax, often leading to sentences that lose sense. There is big room for improvement in this respect since in the current version these errors truly put in risk the scientific clarity. I attach to this review an annotated pdf of the submission to highlight some of these problematic points directly in the text.

In the annotated version the authors may find also a few other comments to the scientific aspects.

[Response]: Thank you for your carefully check. We carefully review the annotated PDF and make the corresponding corrections point by point, we appreciate your annotated PDF in improving the language quality, the detailed revision as follows:

Line16: which strategy?

[Response]: Actually, “this reproductive strategy” means virgin queens usually mates only once before founding a colony. To avoid the misunderstanding, we add more sentences in addressing this in abstracts, the sentences were listed here:

Line 15-16: In Asian honeybees, virgin queens typically mate only once before founding a colony. This behavior is controlled by the queen-released mandibular pheromone (QMP).

Line20: Give a citation that it acts so...

[Response]: Previously, less studies focusing on the stop mating mechanism in Asian honeybees, but it is mainly our findings, to avoid confusion, we changed this sentence into “However, how the queens prevent additional mating remains elusive.” Thank you for the reminds.

Line29: the SSR and EAG experiments did not show that the sensilla narrowly respond to HOB.

At this stage of knowledge, HOB seemed to disrupt ODA detection which does not mean to narrowly respond.

[Response]: Indeed, in EAG assays, drone didn’t show obvious inverse response to HOB, but in SSR, this phenomenon is significant, and the reason for it is not obvious in SSR figures may be due to the low frequency of the spontaneous activity in neuron A naturally, even neuron A’s activity is fully suppressed, the decrease of the spikes in SSR is not obvious in figure. However, the statistic results showed HOB could significantly ($P < 0.001$) reduced neuron A’s spontaneous frequency. And the other evidence point to the fact that HOB and 9-ODA were both functional on the same receptor and same neuron, rather than just block the 9-ODA’s response, which might couldn’t be simply defined as a “antagonist”.

Line30: italics

[Response]: Thank you for your carefully check, the word is corrected.

Line 24-26: Deorphanization of *AcerOr11* in *Xenopus* oocyte system showed 9-ODA induces robust inward (regular) currents, while HOB induces inverse currents in a dose-dependent manner.

Line38-48: This entire paragraph is pretty much redundant and does not bring any information needed for further understanding. There is no need to say that population dynamics in insects are influenced by some factors, and there is no need to speak about ocimene and other semiochemicals, etc. The manuscript can easily start with the second paragraph targeting directly the topic of the study.

What “vegetative diversity on diets” means?

[Response]: That is really a good advice. In the revised manuscript, we have removed the first paragraph about population dynamics in introduction and discussion parts, to make the manuscript more focus.

Line50-54: These two sentences show essentially the same and one sentence with non-redundant information would do the same job.

Moreover, repeated linking of chemical signaling of mating status with the impact on population dynamics is in my view erroneous and occurs on multiple places of the manuscript. The queen should have enough sperm to make the colony functional after the mating flight. To avoid further mating is of course important for the function of the colony but does not directly translate into population dynamics.

[Response]: Thank you for your guidance. We removed the population dynamics contents in this part and throughout the manuscript, to be more rigorous, we less discuss the biological meaning of this “stop mating” behavior in queens is responsible for saving energy for other important tasks such as oviposition.

Line 37-42: A Virgin honeybee queen usually mates only once with several drones, and for the rest of her prolific life, she will not engage in subsequent mating events ^{1,2}. The queen mandibular pheromone (QMP) plays a key role in regulating colony reproduction. QMP directly triggers the mating behavior and provides information about the mating status of queens. It also inhibits the development of worker ovaries by changing relevant gene expression ³.

1. Butler, C. G. The mating behavior of the honeybee (*Apis mellifera* L.). *J. Entomol.* 46, 1-11, doi:10.1111/j.1365-3032.1971.tb00103.x (2009).
2. Gary, N. E. & Marston, J. Mating behavior of drone honey bees with queen models (*Apis mellifera* L.). *Anim. Behav.* 19, 299-304, doi:10.1016/s0003-3472(71)80010-6 (1971).
3. Sandoz, J. C., Deisig, N., de Brito Sanchez, M. G. & Giurfa, M. Understanding the logics of pheromone processing in the honeybee brain: from labeled-lines to across-fiber patterns. *Front. Behav. Neurosci.* 1, 5, doi:10.3389/neuro.08.005.2007 (2007).

Line59: please, provide reference.

[Response]: Thank you for your advice, we added two more references to prove HVA is absent in *A. cerana*.

Line 46-47: In *A. cerana*, 9-ODA, 9-HDA and HOB are also the dominant component of QMP, while HVA is absent ^{4,5}.

4. Plettner, E. *et al.* Species- and caste-determined mandibular gland signals in honeybees (*Apis*). *J. Chem. Ecol.* **23**, 363-377, doi:10.1023/B:JOEC.0000006365.20996.a2 (1997).
5. Keeling, C. I., Otis, G. W., Hadisoelilo, S. & Slessor, K. N. Mandibular gland component analysis

in the head extracts of *Apis cerana* and *Apis nigrocincta*. *Apidologie* **32**, 243-252, doi:10.1051/apido:2001126 (2001).

Line63: please, provide reference

[Response]: We have added more references to support this idea.

Line 50-51: However, the role of HOB remains poorly understood, as single secondary QMP components are not attractive to drones ^{6,7}.

6. Pankiw, T. *et al.* Mandibular gland components of european and africanized honey bee queens (*Apis mellifera* L.). *J. Chem. Ecol.* **22**, 605-615, doi:10.1007/bf02033573 (1996).

7. Keeling, C. I., Slessor, K. N., Higo, H. A. & Winston, M. L. New components of the honey bee (*Apis mellifera* L.) queen retinue pheromone. *Proc. Natl. Acad. Sci. USA* **100**, 4486-4491, doi:10.1073/pnas.0836984100 (2003).

Line68: please, provide references

[Response]: We have added more references to support this idea.

Line 55-56: which expresses the *A. mellifera* odorant receptor 11 gene (*AmelOr11*) ⁸.

8. Kaissling, K. E. & Renner, M. Antennale Rezeptoren für Queen Substance und Sterzelduft bei der Honigbiene. *Zeitschrift für Vergleichende Physiologie* **59**, 357-361, doi:10.1007/bf00365967 (1968).

Line75: QMP composition is not olfactory system

[Response]: Thank you for your careful check. The sentence was corrected into “they differ in their olfactory systems, as well as QMP composition”

Line 61-64: Although *A. mellifera* and *A. cerana* share overall similar morphology and social behavior, they differ in their olfactory systems, as well as QMP composition, the number of odorant receptors (Ors), and antennal lobe topology ¹¹.

11. McKenzie, S. K., Fetter-Pruneda, I., Ruta, V. & Kronauer, D. J. Transcriptomics and neuroanatomy of the clonal raider ant implicate an expanded clade of odorant receptors in chemical communication. *Proc. Natl. Acad. Sci. USA* **113**, 14091-14096, doi:10.1073/pnas.1610800113 (2016).

Line77-80: AcerOR11 is targeted here for some reason, and the reason is that it is orthologous to AmelOR11. This should be made clear here.

[Response]: Thank you for pointing out this problem. We add one sentence here as “Our transcriptome data showed the 9-ODA receptor orthologous is also existed in *A. cerana*, which shed a light on discovery the olfactory pathway of sex pheromone sensing in Asian honey bees.”

Line 65-70: In *A. cerana*, odorant binding protein 11 (AcerOBP11) demonstrates strong binding

affinities for both 9-ODA and HOB¹². However, the specific ORs responsible for detecting 9-ODA and HOB in *A. cerana* remain unclear. Our transcriptome data showed that the 9-ODA receptor ortholog also exists in *A. cerana*, which shed a light on discovery the olfactory pathway on sex pheromone sensing in Asian honey bees.

12. Song, X. M. *et al.* Various Bee Pheromones Binding Affinity, Exclusive Chemosensillar Localization, and Key Amino Acid Sites Reveal the Distinctive Characteristics of Odorant-Binding Protein 11 in the Eastern Honey Bee, *Apis cerana*. *Front. Physiol.* **9**, 422, doi:10.3389/fphys.2018.00422 (2018).

Line81: This should be put into context of the previous findings. Is it new? Or is it notorious? What about other *Apis* species and HOB and mating status?

[Response]: Thank you for the suggestions. The paper you listed truly showed low levels of HOB is detected in virgin queens, however, in most other QMP studies, as well as ours, showing HOB is fully absent in virgin queens. To avoid misunderstand, we corrected this sentence into “In this study, we found that HOB is only released by mated queens, which is consist with most QMP studies [X], and could significantly reduce the attraction of drones to 9-ODA.

Line 71-72: In this study, we found that HOB, released only by mated queens, significantly reduces the attraction of drones to 9-ODA.

Line83: electroantennography

[Response]: Thank you for the check, corrected throughout the manuscript.

Line 72-74: his inverse effect of HOB was further validated by *in vivo* electrophysiological assays, electroantennography (EAG), and single sensillum recording (SSR).

Line88-89: redundant.

[Response]: It was deleted.

Line 77-78: This study aimed to explore how QMP regulates reproduction at the olfactory sensing level of *A. cerana*.

Line105: two columns are listed. What was the separation column?

[Response]: Thank you for your reminder. BSTFA is the short name for the “N,O-Bis (trimethylsilyl) trifluoroacetamide”, to avoid any misunderstands, the “(BSTFA)” was removed.

Line 92-94: The extracts were then dried under a stream of nitrogen, and the remainders were dissolved in 20 µL internal standard solution (octanoic acid and tetradecane in dichloromethane) and 20 µL N,O-

Bis (trimethylsilyl) trifluoroacetamide.

Line110: to

[Response]: Corrected.

Line 99-100: Helium gas was used as a carrier gas at a constant flow rate of 1 mL/min. The oven temperature was set to 100 °C for 2 min and then increased to 250 °C at a rate of 10 °C per minute.

Line113: I assume that using GCMS also allowed to identify the compounds based on their mass fragmentation.

[Response]: Thank you for your carefully check, indeed, both retention times and fragmentation patterns were used in identifying target compounds, the word “mass fragmentation” was added to the sentence.

Line 100-102: The final temperature was maintained for 10 min. The compounds were identified by comparing their retention times and mass fragmentation with the known reference compounds.

Line117: please, add “solution” or “dilution”

[Response]: Thank you for the check, the word “solution” was added.

Line 106-108: Briefly, 10 µL of a test compound solution was applied to a 25 × 15 mm filter paper and then placed in one arm of the olfactometer (15 cm base, 10 cm arm length, and 2 cm diameter) as the odorant source.

Line122: plural

[Response]: “concentration” was corrected into “concentrations” in revised version.

Line 110-112: More than thirty honeybees were used in each behavioral assay for a series of test compound concentrations.

Line124: both ends of the antenna? Or ends of both antennae? Please clarify here.

[Response]: The word was corrected into “The tip of the antennae”.

Line 114-116: In the EAG assay, the tip of the honeybee antennae were cut and covered with a conductive gel (Parker Laboratories Inc., USA), and the honeybee head was attached to the reference electrode.

Line149: plural

[Response]: Corrected.

Line 139-140: The two antennae were exposed to a stimulus for 500 ms with airflow of 0.6 L/min

through a Pasteur pipette.

Line153-154: rather “subtraction of the spontaneous spikes FROM the firing spike numbers.”

[Response]: “of” was corrected into “from” instead.

Line 142-144: The number of induced spikes were calculated as the subtraction from the firing spike number by spontaneous spikes number before the stimulus.

Line162: which caste?

[Response]: Sorry for misleading, three castes were all included, we add more details here.

Line 152-154: Antennae of 12- to 15-day-old honeybees of three castes were collected and then embedded in a Tissue-Tek optimal cutting temperature compound (Sakura Finetek, USA).

Line224: analyzed

[Response]: Corrected

Line 214-215: Data collected in TEVC were analyzed by Clampfit 10 software.

Line231: contents

[Response]: Corrected

Line 221-222: To measure the contents of QMPs in *A. cerana*, we performed GC-MS analysis on 12- to 15-day-old virgin queens, mated queens as well as drones and workers.

Line233: plural

[Response]: Corrected

Line 222-224: 9-ODA and 9-HDA, two main components in QMPs, were detectable in both virgin and mated queens (Figure 1A, B, S1).

Line234: “mated queens other than virgin”? Please, reword.

[Response]: “HOB was only detected in mated queens other than virgin” was corrected into “HOB was only detected in the mated queens and not in virgin queens”.

Line 224-225: However, interestingly, HOB was only detected in the mated queens and not in virgin queens (Figure 1B).

Line235: please reword

[Response]: Changed into “inspired us”.

Line 225-226: This inspired us that HOB might exercise some functions on the post-mating regulation.

Line238: past tense

[Response]: “aim” was changed into “aimed”

Line 228-229: Thus, we aimed to test the behavioral effects of HOB on drones, using the Y-tube olfactometer assay.

Line238: “drones. We” Please, split in two sentences or reword.

[Response]: The two sentences were reworded.

Line 228-229: Thus, we aimed to test the behavioral effects of HOB on drones, using the Y-tube olfactometer assay.

Line240: please remove “were” or exchange “at” for “if”

[Response]: Corrected.

Line 229-231: First, we used 9-ODA as a stimulus and found that it has a significant attractive effect on drones only at the highest concentration (100 µg) ($p < 0.05$, two-tailed, T-test).

Line244: doses

[Response]: Corrected.

Line 234-235: When we mixed 100 µg of 9-ODA with different concentrations of HOB (0.1-100 µg)

Line245: was

[Response]: Corrected

Line 235-236: we found that 9-ODA’s attraction was suppressed by HOB at concentration beyond 1 µg.

Line256: “in any caste”

[Response]: We gratefully appreciate for your valuable suggestion. We have corrected in revised manuscript.

Line 246-247: On the contrary, HOB did not elicit any significant antennal response in any castes (Figure 3B and S2).

Line262: replace with “and”

[Response]: Thank for your careful checks. We have replaced it.

Line 253-254: and the effects were significantly lower than those from 9-ODA alone (Figure 3D).

Line266: Please, delete and start the sentence as “Three types...”

[Response]: We sincerely thank you for your careful reading. We have corrected it.

Line 258-260: Three types of chemosensory sensilla were observed in the *A. cerana* drone’s antenna, including the sensilla trichodea, placodea, and basiconica.

Line277: neither...nor...changed

[Response]: Thank you for your nice suggestion. We will pay more attention.

Line 269-270: Intriguingly, neither 9-ODA nor HOB changed the spontaneous activity of B and C neuron (Figure 4D).

Line283: implicating that

[Response]: We are very sorry for our negligence, and we have corrected in revised manuscript.

Line 275-276: Overall, these results implicating that HOB could act as an inverse agonist at the ORNs level.

Line285: quantifying AcerOR11 in different tissues does not help understanding “the potential molecular mechanism”.

[Response]: This sentence was corrected into “To check the expression profile of *AcerOr11*.”

Line 278: To check the expression profile of *AcerOr11* underlying the physiological response...

Line288: So, is it expressed only in the drone antenna or little expressions were also detected elsewhere?

[Response]: The word “only” was corrected into “abundantly”

Line 280-282: We found that *AcerOr11* is abundantly expressed in drone antennae, while little expression was also detected in the queen and worker antennae (Figure S4).

Line289: This is a very good hint the AcerOR11 is important but not a proof yet. The proof (and very convincing proof) will only come later in this chapter

[Response]: This sentence was moved to the end of this chapter.

Line293: “bees from the castes”? Please, reword.

[Response]: Corrected.

Line 284-285: The DIG-labeled riboprobes for *AcerOr11* were applied to transversal antennal sections of bees from three castes.

Line295: does not make much sense, please, reword.

[Response]: The sentence was corrected.

Line 287-289: Only a few cells in workers' antennae expressed *AcerOr11*, while *AcerOr11* was fully undetectable in queens' antennae (Figure 5C, D, L, M).

Line298: areas

[Response]: Corrected.

Line 290-291: *AcerOr11*-labeled areas were mainly distributed in dendrite-like structures of ORNs housed in sensilla placodea (Figure 5K).

Line306: singular

[Response]: Corrected.

Line 297-298: We functionally expressed *AcerOr11* in the *Xenopus laevis* system and screened it with a 163-compound panel.

Line310: response of AerOR11 to HOB, not the opposite.

[Response]: It is true as you suggested that it should be the “response of *AcerOr11* to HOB”, and we have corrected in revised manuscript.

Line 302-303: To further confirm the inverse response of *AcerOr11* to HOB, a current-voltage (I-V) curve was constructed using 9-ODA and HOB as stimuli.

Line320: please, reword

[Response]: Reword.

Line 312-313: We fixed the concentration of 9-ODA to 10^{-5} M for a saturated response while increasing the HOB concentration from 10^{-6} M to 10^{-3} M.

Line327: series

[Response]: Corrected.

Line 318-320: Subsequently, we used 9-ODA alone for a series of concentrations tests and found that the response to 9-ODA was recovered (Figure 6G).

Line334: references

[Response]: We have added reference to support this viewpoint. Thank you for your reminder.

Line 326-329: HOB, a pheromone signal only released by mated queens, exerts a “stop mating” function on drones. While a few studies have also reported the presence of tiny levels of HOB in virgin queens 13, most of the QMP studies, as well as ours, showed HOB is fully absent in virgin queens

4,14.

4. Plettner, E. et al. Species- and caste-determined mandibular gland signals in honeybees (*Apis*). *J. Chem. Ecol.* 23, 363-377, doi:10.1023/B:JOEC.0000006365.20996.a2 (1997).

13. Strauss, K. et al. The role of the queen mandibular gland pheromone in honeybees (*Apis mellifera*): honest signal or suppressive agent? *Behav. Ecol. Sociobiol.* 62, 1523-1531, doi:10.1007/s00265-008-0581-9 (2008).

14. Villar, G., Hefetz, A. & Grozinger, C. M. Evaluating the Effect of Honey Bee (*Apis mellifera*) Queen Reproductive State on Pheromone-Mediated Interactions with Male Drone Bees. *J. Chem. Ecol.* 45, 588-597, doi:10.1007/s10886-019-01086-0 (2019).

Line344: Again, I do not see a causal link here to claim this.

[Response]: The sentence was corrected.

Line 338-340: Our behavioral assay results indicate that HOB directly inhibits the mating behavior of drones and might prevent the queens from consuming the energy for redundant mating.

Line349: please, provide a correct name for the compound.

[Response]: Thanks for your careful checks. We have read the reference and correct its name.

Line 342-343: In *Helicoverpa armigera*, female moths release cis-11-hexadecenol (Z11-16: OH) to repel males and avoid non-optimal mating¹⁶.

16. Chang, H. et al. A Pheromone Antagonist Regulates Optimal Mating Time in the Moth *Helicoverpa armigera*. *Curr. Biol.* 27, 1610-1615, doi:10.1016/j.cub.2017.04.035 (2017).

Line352: their.

[Response]: Corrected.

Line 346: With a few exceptions, honeybee queens do not remate in their lifetime.

Line360: please, provide genus name upon first occurrence

[Response]: Thank you for your reminder. We have updated the full name of *Culex quinquefasciatus*.

Line 354-356: In *Culex quinquefasciatus*, eucalyptol significantly decreases the number of spikes in the ab7 sensillum.

Line363: unclear, please reword

[Response]: The sentence was corrected.

Line 356-358: Besides the interreceptor inhibition, here, we show that HOB reduces the activity of 9-ODA at the same ORNs, supporting the hypothesis of the intrareceptor inhibition.

Line367: acts

[Response]: Corrected.

Line 361-362: We speculate that HOB acts as an inverse agonist to modulate mating behavior at the ORNs level.

Line486: here, you can show the retention times of the synthetic standards in a separate chromatogram, or provide some spectra in supplementary.

[Response]: Thank you for your valuable suggestion. As you suggested, we have supplied the retention times of the synthetic standards (9-ODA and HOB) in Figure S1 in “Supplementary materials”.

[Reviewer 3 major comments]:

Reviewer #3 (Remarks to the Author):

This manuscript describes experiments that identify HOB, a substance of the queen mandibular pheromone, as an anti-aphrodisiac in *Apis cerana*. The interesting result is that HOB seems to inhibit the sex pheromone 9-ODA, that usually attracts males, by directly affecting OR11, the odorant receptor tuned to 9-ODA. I found the results very interesting and did not find any experimental or conceptual flaws, although I have never myself worked on odorant perception on the molecular level. What I did not enjoy was the general presentation of the manuscript. Throughout I was wondering what the biological function of the mechanism described here could be. In the discussion the authors then speculate that an anti-aphrodisiac could reduce mating in the colony, which would reduce colony growth. Either I am misunderstanding something or this does not make any sense. The queens do not mate in the colony, and matings do not have any direct effects on egg-laying rates (other than that the queen needs to be mated at least once to lay fertilized eggs). The language requires some editing overall, there are many errors (e.g. mixed plural & singular, inconsistent use of past and present tense, typos), and a few more details on the exact procedures would be nice as well (see below).

[Response]: Thank you for valuable comments on our manuscript. We have paying particular attention to enhancing the English language and grammar accuracy throughout the revised manuscript. Meanwhile, to be more rigorous, we removed the population dynamic introduction and discussion in the revised version, instead, we just mentioned this “stop mating” behavior as an energy saving strategy in queens to be a trade-off to their other important tasks such as oviposition. Our detailed responses were as follows:

line 16 “this strategy” which strategy?

[Response]: Sorry for the confusion. Actually, “this reproductive strategy” means virgin queens usually mates only once before founding a colony. To avoid the confusion, we rewrite this part in abstract, the details are as follows:

Line 15-16: In Asian honeybees, virgin queens typically mate only once before founding a colony. This behavior is controlled by the queen-released mandibular pheromone (QMP).

20 why would one expect that qmp prevents mating?

[Response]: Previously, few studies are focusing on the stop mating mechanism in Asian honeybees, but it is mainly our findings, to avoid confusion, we changed this sentence into “However, how the queens prevent additional mating remains elusive.” Thank you for the reminds.

38 I don't see the connection to population dynamics at all, in particular regarding the environmental factors mentioned in the first sentence

[Response]: Thank you for your professional advice. The reproduction and the maintain of sociality in honeybee colony are intricately linked to chemical signals. However, the regulate of population dynamics are influenced by a variety of factors instead of chemical signals simply. Your suggestion has greatly inspired us. Consequently, in the revised manuscript, we have excised the content pertaining to population dynamics and instead augmented it with more comprehensive discussions on chemical signals between mating behavior, to solidify the foundation for our research.

Is AcerOr11 orthologous to AmelOR11?

[Response]: Thank you for pointing out this problem. We add one sentence here as "Our transcriptome data showed the 9-ODA receptor orthologous is also existed in *A. cerana*, which shed a light on discovery the olfactory pathway of sex pheromone sensing in Asian honey bees."

Line 65-70: In *A. cerana*, odorant binding protein 11 (AcerOBP11) demonstrates strong binding affinities for both 9-ODA and HOB¹². However, the specific ORs responsible for detecting 9-ODA and HOB in *A. cerana* remain unclear. Our transcriptome data showed that the 9-ODA receptor ortholog also exists in *A. cerana*, which shed a light on discovery the olfactory pathway on sex pheromone sensing in Asian honey bees.

12. Song, X. M. *et al.* Various Bee Pheromones Binding Affinity, Exclusive Chemosensillar Localization, and Key Amino Acid Sites Reveal the Distinctive Characteristics of Odorant-Binding Protein 11 in the Eastern Honey Bee, *Apis cerana*. *Front. Physiol.* **9**, 422, doi:10.3389/fphys.2018.00422 (2018).

182 be more specific about the tissues. why this choice? You say including, is this the full list of tissues? (later it becomes clear that tissues were analysed separately to locate the expression of the genes in question)

[Response]: We gratefully appreciate for your valuable comment. In fact, we tested AcerOr11 expression level in chemosensory organs and non-chemosensory body parts (as a negative control). To make it clearer, we rewrite the corelated sentences in revised manuscript.

Line 172-174: RNA samples from different tissues, including the chemosensory organs (antenna, proboscis) and non-chemosensory body parts (thorax, abdomen, and legs), were collected from 15 bees per caste.

231 countents typo

[Response]: Corrected

Line 221-222: To measure the contents of QMPs in *A. cerana*, we performed GC-MS analysis on 12- to 15-day-old virgin queens, mated queens as well as drones and workers.

Fig 1: It would be nice to learn more about sample size etc. here. I find it a bit weird that the data in table S3 indicate that there was never 9-ODA or HOB in workers but in Fig. 1 there are clearly small peaks.

[Response]: Thank you for pointing out this problem in manuscript. In GC-MS analysis, we conducted three biological replications. Although we can see a tiny peak in workers, the contents of it is very small compared to that of queens, it is nearly undistinguishable from the background noise and even hardly detectable in the GC-MS analysis. Meanwhile, in the other two biological replicates, 9-ODA and HOB are fully absent in works. In summary, considering these results, we concluded that 9-ODA and HOB are not released by workers, or to be more rigorous, they were so tiny that even undetectable by GC-MS. Thank you for your suggestions again.

241 “at as”?

[Response]: Corrected.

Line 229-231: First, we used 9-ODA as a stimulus and found that it has a significant attractive effect on drones only at the highest concentration (100 μg) ($p < 0.05$, two-tailed, T-test).

242, 246 I would give separate p-values for the different concentrations in the text

[Response]: Thank you for your comment. We have list p values for different concentrations in revised version.

Line 231-233: At a lower concentration of 0.1-10 μg , the attraction effect of 9-ODA was not significant ($p = 0.5614$ for 0.1 μg ; $p = 0.9580$ for 1 μg ; $p = 0.0668$ for 10 μg) (Figure 2A).

Line 234-237: When we mixed 100 μg of 9-ODA with different concentrations of HOB (0.1-100 μg), we found that 9-ODA's attraction was suppressed by HOB at concentration beyond 1 μg ($p = 0.2302$ for 1 μg ; $p = 0.3771$ for 10 μg ; $p = 0.9414$ for 100 μg) (Figure 2C).

Fig. 4F I don't understand how there can be negative spikes per second. Is this compared to some sort of baseline of the same antenna? When is the baseline measured? Please indicate this in the figure

[Response]: We totally understand your concern. The number of induced spikes were calculated by the firing spike number subtraction from spontaneous spikes number before the stimulus. This information was added in the relevant figure.

301 is it really expressed in the cytoplasm or is the RNA quickly moved out of the nucleus after transcription?

[Response]: Thank you for your rigorous comment. Insect OR is a family belongs to transmembrane receptors families, so the mRNA of ORs should be quickly moved from the nucleus to cytoplasm after transcription.

327 "serious"?

[Response]: Corrected.

Line 318-320: Subsequently, we used 9-ODA alone for a series of concentrations tests and found that the response to 9-ODA was recovered (Figure 6G).

343 "results indicate that HOB directly inhibits the mating behavior of drones and might prevents the bee colony from having an oversized population, trading off the number of offspring for the allocation

of energy resources, including physical resources and parental care.”

I'm not a bee person but that sounds very wrong. The drones do not mate the queen inside the colony. The queen typically takes one mating flight and then lays eggs continuously, independent of further matings. As far as I know this is also true for *Apis cerana*. Now I understand the weird beginning of the introduction talking about colony dynamics. It does not seem to make sense to me.

[Response]: To be more rigorous, we removed the population dynamic discussion in the revised version, instead, we changed this sentence into “results indicate that HOB directly inhibits the mating behavior of drones and might prevents the queens from consuming the energy for redundant mating.”

398 This last paragraph is weird. The findings are interesting in themselves, no need to invoke spurious arguments about conservation. If you invoke conservation, explain how you would use HOB to conserve *A. cerana*.

[Response]: We gratefully thanks for your constructive remarks. As your suggested, we deleted this paragraph in the revised manuscript.

REVIEWERS' COMMENTS:

Reviewer #1 (Remarks to the Author):

I congratulate the authors for a systematic revision of their manuscript according to the reviewer's suggestions. I think that the revisions improved substantially the manuscript, which is now more suitable for publication in Commun Biol.

I attach to this review an annotated text file of the revision, in which I identify some additional rather minor modifications to be made by the authors. Please, see my correction in „track change“ mode and a few comments inserted into the text.

[Reviewer 1 major comments]:

Reviewer #1 (Remarks to the Author):

Line15: This may seem that they are monogamous. You can write something like “virgin queens only mate during a single mating flight”.

[Response]: We have corrected, thank you for your suggestion.

Line15-16: In Asian honeybees, virgin queens typically only mate during a single nuptial flight before founding a colony.

Line68: Which data? Provide a citation or say that it is unpublished data.

[Response]: We have added reference, thank you.

Line67-69: Our transcriptome data showed that the 9-ODA receptor ortholog also exists in *A. cerana*, which shed a light on discovery the olfactory pathway on sex pheromone sensing in Asian honey bees¹³.

¹³ Ke, H. *et al.* Odorant Receptors Expressing and Antennal Lobes Architecture Are Linked to Caste Dimorphism in Asian Honeybee, *Apis cerana* (Hymenoptera: Apidae). *Int. J. Mol. Sci.* **25**, doi:10.3390/ijms25073934 (2024).

Line280/296: Please, decide whether you will call AcerOR11 a homolog or an ortholog of AmelOR11 based on phylogenetic patterns.

[Response]: As you said, the OR11 clade also included OR12, they were all homologs, and to be more accurate, OR11 should be called “ortholog”, we corrected this, thank you for the check.

Line137-138: we conducted an RT-qPCR survey of the *AmelOr11* ortholog in *A. cerana* in different tissues and castes.

Line153-154: To identify the QMP ORs in *A. cerana*, we cloned *AcerOr11*, which is the 1:1 ortholog of *AmelOr11*...

Line337-340: This still seems redundant to me. It simply acts a “stop mating” with a biological function clear to me.

[Response]: Thank you for your advice. We have corrected “population dynamics” into “stop mating”, thank you again.

Line194-195: and might play a crucial role in stop mating in honeybees.

Line343: (Z)-11-Hexadecen-1-ol

[Response]: Corrected.

Line199-200: In *Helicoverpa armigera*, female moths release (Z)-11-Hexadecen-1-ol (Z11-16: OH) to repel males and avoid non-optimal mating.

Line348-349: Again, I do not feel it is correct. The queen simply needs to be ignored by drones once she is egg-laying. A speculative adaptive significance may be, for instance, that she needs to be ignored by drones emerging (developing) in her own nest... the view that the HOB has an impact population dynamic is wierd.

[Response]: We remove the sentence about describe over-sized populations. Thank you.

Line204-205: Mated queens release HOB to prevent drones from multiple mating, which can help in avoiding resource waste due to multiple mating and subsequent breeding.